# Subversion of phosphorylated SR proteins by enterovirus A71 in IRES-dependent translation revealed by RNA-interactome analysis

Kuo-Ming Lee[1,2,3]*, Chih-Ching Wu[2,4,5,6], Yu-Ting Fan[1], Huan-Jung Chiang[5], Pei-Yi Lien[2], Jui-Ping Wang[4], Yhu-Chering Huang[3], Shin-Ru Shih[2,4,7]

**1** International Master Degree Program for Molecular Medicine in Emerging Viral Infections, College of Medicine, Chang Gung University, Taoyuan, Taiwan, **2** Research Center for Emerging Viral Infections, College of Medicine, Chang Gung University, Taoyuan, Taiwan, **3** Division of Pediatric Infectious Diseases, Chang Gung Memorial Hospital, Linkou, Taoyuan, Taiwan, **4** Department of Medical Biotechnology and Laboratory Science, College of Medicine, Chang Gung University, Taoyuan, Taiwan, **5** Graduate Institute of Biomedical Sciences, College of Medicine, Chang Gung University, Taoyuan, Taiwan, **6** Department of Otolaryngology-Head & Neck Surgery, Chang Gung Memorial Hospital, Linkou, Taoyuan, Taiwan, **7** Emerging Infectious Disease Division, Biomedical Translation Research Center & Institute of Biomedical Sciences, Academia Sinica, Taipei, Taiwan

* g39002007@mail.cgu.edu.tw

## Abstract

During infection by positive-sense single-stranded RNA viruses, understanding the mechanisms governing the fate of viral RNA, whether directed towards translation, replication, or virion assembly, remains a significant challenge. In this study, we conducted RNA-interactome analysis using metabolic labeling coupled with quantitative proteomics to investigate the protein composition of temporal ribonucleoprotein complexes (RNPs) during enterovirus A71 (EV-A71) infection. Comparative analysis of RNPs during the early and late infection stages, representing the eclipse and maturation phases, revealed dynamic RNP remodeling over time. This remodeling process involved the exchange of nuclear RNA binding proteins with cytoplasmic membrane-associated proteins. Notably, EV-A71 infection induced the phosphorylation and cytoplasmic re-localization of nuclear serine and arginine-rich (SR) proteins, as determined using pan-SR protein antibodies, with these proteins found to co-localize with viral RNAs. Knockdown of specific SR proteins, including SRSF4, SRSF5, and SRSF6, significantly reduced viral growth, highlighting their critical role in the infection process. Intriguingly, these phosphorylated SR proteins cofractionated with the translation machinery rather than the replication organelles, a phenomenon predominantly observed during the early infection phase and abolished in the late phase. Importantly, inhibition of SR protein phosphorylation using the kinase inhibitors SRPKIN-1 and TG003 significantly impaired IRES-dependent translation and EV-A71 replication. These findings underscore the pivotal role of SR protein phosphoregulation during the eclipse phase of EV-A71 infection in facilitating the formation of translation-competent complexes. Furthermore, they highlight

**Data availability statement:** All data generated or analyzed during this study are included in the published article and its Supporting Information files. The protein identification data are provided in the supplemental tables (S1–S3 Tables). The raw mass spectrometry data have been deposited in the PRoteomics IDEntifications Database (PRIDE) and are publicly available under the accession number PXD036890 (https://www.ebi.ac.uk/pride/archive/projects/PXD036890).

**Funding:** This work was financially supported by the National Science and Technology Council (NSTC) [formerly known as the Ministry of Science and Technology (MOST)], Taiwan (MOST 109-2320-B-182-045-MY2 and NSTC 112-2320-B-182-043-MY3 to K-ML; MOST 111-2321-B-182-001 to S-RS), and by the Ministry of Education (MOE), Taiwan, through the Featured Areas Research Center Program within the framework of the Higher Education Sprout Project (no grant number) to the Research Center for Emerging Viral Infections, Chang Gung University. The funders had no role in study design, data collection and analysis, decision to publish, or preparation of the manuscript.

**Competing interests:** The authors have declared that no competing interests exist.

the potential of targeting SR protein phosphorylation as a promising strategy for antiviral development.

## Author summary

Since the 1990s, periodic outbreaks of Enterovirus-A71 (EV-A71) have posed a significant health threat in the Asia-Pacific region, occasionally leading to acute flaccid paralysis. Some outbreaks have been linked to the emergence of novel recombinant strains, underscoring the pressing need for antiviral development and a focus on conserved mechanisms to mitigate drug resistance attributed to recombination and mutation. To gain insight into how the viral genome can serve distinct roles in translation, replication and virion assembly during EV-A71 infection, we employed 4-thiouridine tagging to label newly synthesized RNA, facilitating the isolation of temporal ribonucleoprotein complexes (RNPs) for compositional analysis. Our investigation revealed that during the early replication stage, these RNPs were enriched with certain SR proteins, particularly in their phosphorylated forms. Notably, these virus-induced phosphorylated SR proteins were predominantly associated with translation complexes rather than replication organelles. Moreover, attenuation of both the protein levels and phosphorylation of several SR proteins significantly impeded EV-A71 replication, as well as that of other enterovirus species. Thus, our findings offer new targets for broad-spectrum intervention strategies against a range of enteroviruses through a comprehensive understanding of virus-host interactions.

## Introduction

Virus replication relies on intricate virus-host interactions, with positive-sense (+) single-stranded RNA (ssRNA) viruses being a key group of interest due to their distinctive replication cycle. These viruses carry an infectious RNA genome and primarily replicate in the cytoplasm [1]. Genome replication occurs within membranous replication organelles derived from cellular organelles, with the specific structures varying by virus [2]. Central to this process is the viral RNA-dependent RNA polymerase (RdRp), which exhibits a structurally conserved catalytic center despite coding sequence diversity [3]. This conserved mechanism has made RdRp a promising target for antiviral therapies; for instance, ribavirin, a guanosine nucleoside analog, inhibits replication in various viruses by inducing mutation catastrophes [4]. However, the high mutation rate of RNA viruses can lead to the emergence of drug-resistant variants. Therefore, investigating conserved mechanisms that rely on virus-host interactions provides valuable insights for the development of broad-spectrum antivirals.

The replication cycle of (+) ssRNA viruses can be divided into distinct stages: adsorption, eclipse, maturation, and egress. During the eclipse phase, following

receptor binding and genome uncoating, viral RNA serves a dual purpose as both a template for translation and replication. This dual functionality requires precise regulation to synthesize non-structural proteins necessary for genome replication and structural proteins for virion assembly. Consequently, the genomic RNA of (+) ssRNA viruses, such as picornaviruses, must undergo efficient translation immediately after uncoating to ensure a productive infection. However, the (+) RNA genome alternates between serving as a template for translation and replication, a process that must be tightly regulated. Different mechanisms have been proposed to regulate the transition between translation and replication [5–7]. A molecular switch mechanism has been proposed for the proteolytic cleavage of the poly (rC)-binding protein 2 (PCBP2) by enterovirus proteases [8]. PCBP2 interacts with the cloverleaf stem-loop and the internal ribosomal entry site (IRES) within the 5' untranslated region (5' UTR) to regulate poliovirus replication and translation by associating with PABP and serine/arginine-rich splicing factor 3 (SRSF3, formerly known as SRp20) [9–11]. Cleavage of PCBP2 by the poliovirus 3CD protease disrupts its interaction with SRSF3, thereby inhibiting viral translation while leaving replication unaffected [8]. In the context of hepatitis C virus, another mechanism fine-tunes genome usage in replication and translation through PCBP2 competition. This occurs via assembly of an unconventional argonaute RISC catalytic component 2: microRNA-122 (miR-122) complex at the 5' end [12]. The binding of miR-122 prevents the PCBP2 binding, opening the genome structure for replicase assembly at the 3' end [7]. Such spatiotemporal ribonucleoprotein complex (RNP) remodeling resembles host transcript processing.

The maturation of messenger RNA (mRNA) involves various ribonucleoprotein complexes (RNPs), such as the capping enzyme, 3' end processing complex, and spliceosome. These RNPs assembled sequentially along the elongating RNA polymerase II (Pol II) during pre-mRNA processing. This stepwise assembly not only enhances processing efficiency but also serves as a quality control mechanism to ensure that only fully processed and functional mRNAs are expressed [13,14]. Similarly, during viral infection, distinct viral RNPs (vRNPs) are formed to carry out translational and replicational roles. Therefore, the aforementioned PCBP2 cleavage favors replication by inhibiting the formation of a translation-competent complex. This occurs through the disruption of PCBP2's association with SRSF3, an essential IRES trans-acting factor (ITAF) required for ribosome assembly [8,11]. Additionally, the 2A protease of several enteroviruses induces the cytoplasmic translocation of SRSF3, resulting in stress granules-like cytoplasmic foci that may regulate viral translation during mid to late infection [15,16]. Another example is the competitive or additive effect between far upstream element binding protein 1 (FUBP1), KH-type splicing regulatory protein (KHSRP), and their proteolytic products on enterovirus A71 (EV-A71) IRES-dependent translation [17]. Moreover, (+) RNA serves a crucial role in encapsidization for packaging the genome into progeny virions. However, the detailed composition of vRNPs with different functions remains unclear and can be elucidated through RNA interactome analysis. This technique, based on ultraviolet (UV) cross-linking of infected cells, covalently links proteins in close proximity (~2 Å distance) to RNA [18,19]. Recent studies have revealed steady-state RNP components of various viruses, including poliovirus, dengue virus, chikungunya virus, influenza virus, and SARS-CoV-2 [20–25].

Herein, we focused on EV-A71, a non-polio enterovirus known for causing paralysis [26,27]. Since the late 1990s, periodic EV-A71 outbreaks have occurred in the Asia-Pacific region, particularly in Taiwan, where EV-A71 stands as the primary enterovirus associated with severe neurological complications. Ongoing recombination with other enteroviruses gives rise to novel strains, raising concerns about future EV-A71 pandemics [28]. Despite extensive efforts in the development of EV-A71 vaccines and antivirals targeting RdRp and IRES [29–31], none have gained approval thus far. We previously characterized the IRES-dependent EV-A71 translation, uncovering multiple regulatory factors and mechanisms, including ITAFs, small RNAs, and proteolytic cleavage [17]. To deepen our understanding of EV-A71, we conducted an RNA interactome study via utilizing metabolic labeling of infected cells with 4-thiouridine (4sU). The membrane-permeable 4sU can be specifically activated by 365-nm UV light [18,19], thereby reducing the non-specific binding. Through combined quantitative proteomics, we explored the components of RNPs specifically isolated during the early and late infections, characterizing their functional impacts on the virus replication.

## Results

### Metabolic labeling of replicating RNAs of EV-A71 by host-transcription shutoff

To enhance the proportion of viral RNAs (vRNAs) within the pool of labeled transcripts during the eclipse phase, we inhibited host transcription using actinomycin D (ActD), 5,6-Dichlorobenzimidazole 1-β-D-ribofuranoside (DRB), and α-amanitin, each targeting distinct spectra of host polymerases. The effective concentrations used were: 0.05 μg/ml (0.04 μM), 0.5 μg/ml (0.4 μM), and 5 μg/ml (4 μM) of ActD to inhibit RNA Pol I, II, and III, respectively; 100 μM DRB for RNA Pol II; and 2 μg/ml (2 μM) α-amanitin for RNA Pol II [32]. The inhibitory effects of each compound at its highest recommended concentration were confirmed in human rhabdomyosarcoma (RD) cells using reverse transcription quantitative polymerase chain reaction (RT-qPCR). All inhibitors were maintained in the culture medium throughout the entire experimental period. The results demonstrated that the mRNA levels of the four genes examined decreased to less than 50% after 2 hours (h) of drug treatment, with drug ActD exhibiting the most pronounced effect (S1A Fig). Subsequently, we evaluated their impact on viral replication, and RD cells infected with EV-A71 were treated with each inhibitor for 12 h. Both ActD and DRB significantly reduced viral titers, whereas the RNA Pol II-specific inhibitor, α-amanitin, showed no effect (Fig 1A). The insensitivity of EV-A71 to α-amanitin was further confirmed by the similar growth kinetics and vRNA synthesis (Fig 1B and 1C). Next, we conducted the metabolic labeling by feeding 4sU to EV-A71-infected cells for 4 and 8 h, representing the eclipse and maturation phases, respectively (Fig 1D). Total RNA was recovered, and the incorporation of 4sU was confined to newly synthesized RNAs, which underwent thiol-specific biotinylation followed by pull-down assays using streptavidin beads. Reverse transcription polymerase chain reaction (RT-PCR) analysis of the selected RNAs demonstrated specific labeling of both host and viral transcripts by 4sU (Fig 1D, compare lanes 4–6 and 10–12). We further quantitatively assessed the impact of α-amanitin on the pulldown efficiency of viral and host RNAs using qPCR analysis in a representative experiment. The results showed that α-amanitin treatment improved the pulldown efficiency of viral RNA at 4 h post-infection (hpi) from 12.2% to 23.9%, comparable to the levels of viral transcripts observed at 8 hpi in the non-treatment group (24.9%) (Fig 1E, IRES panel). Furthermore, the pulldown efficiency of viral transcripts was similar to that of the highly abundant housekeeping genes from the host (Fig 1E). Taken together, these results confirm that the successful metabolic labeling of EV-A71-infected cells enabled us to isolate RNAs from different time points of infection.

Next, we utilized UV crosslinking to investigate the temporal dynamics of RNP components throughout the infectious cycle of EV-A71. Infected cells were cultured in medium containing α-amanitin, with or without 4sU. After 4 or 8 h, cells were exposed to UV light at 365 nm, specifically activating the thiol group of labeled RNAs to crosslink with nearby proteins. RNPs were then purified using oligo(dT) magnetic beads (Fig 1F, upper schematic). To confirm the specific enrichment of poly(A) RNA by oligo(dT) pulldown, we compared the recovery efficiency of polyadenylated transcripts with that of 18S ribosomal RNA (rRNA). RT-qPCR analysis revealed that more than 10% of viral RNA (19.2% and 12.9% for the 4-h and 8-h RNPs, respectively) and approximately 10% of host *vimentin* (*VIM*) transcripts were enriched in the pull-down samples. In contrast, less than 5% of the abundant 18S rRNA was recovered, indicating successful and selective enrichment of polyadenylated transcripts, including viral RNA (Fig 1F, lower panel). Notably, the pulldown efficiency of host transcripts varied in a gene-specific manner, likely reflecting differences in RNA stability and abundance following transcriptional inhibition (S1B Fig). To validate the specificity of RNPs isolated at 4 and 8 hpi, we performed western blotting, and detected cellular proteins known to participate in enterovirus translation and replication. ITAF proteins such as FUBP1, KHSRP, and polypyrimidine tract binding protein 1 (PTBP1) were predominantly identified in the 4-h RNPs (Fig 1G, lanes 5 and 7) [17]. In contrast, proteins involved in genome replication were detected in the 8-h RNPs including the 3D$^{pol}$ and hnRNPC (Fig 1G, lanes 6 and 8) [33]. While hnRNPC1/C2 is present at both time points, its prominent abundance at 4 hpi may indicate additional roles beyond replication, such as in IRES-dependent translation regulation [34,35]. These results support the existence of temporally distinct RNPs, with the differential distribution of proteins potentially reflecting stage-specific roles in translation or replication. Notably, the high background of viral proteins in the -4sU condition likely results from their abundant expression at 8 hpi, which may lead to nonspecific binding during oligo(dT)

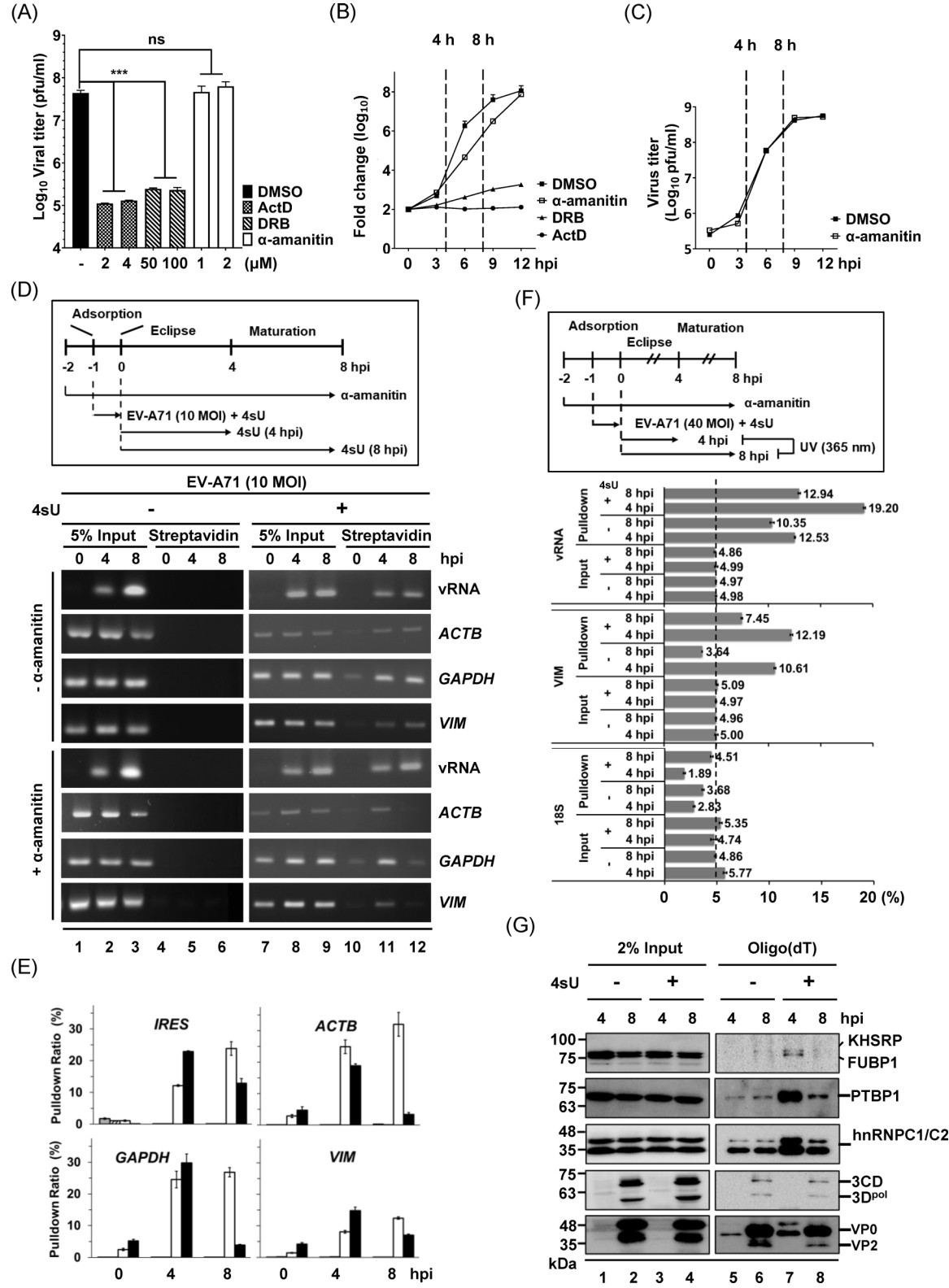

**Fig 1. Establishment of Metabolic Labeling in EV-A71-Infected Cells.** (A) Viral titers of rhabdomyosarcoma (RD) cells infected with EV-A71 at a multiplicity of infection (MOI) of 10 for 12 h in the presence of various concentrations of transcription inhibitors. Each inhibitor was added during viral adsorption and remained in the medium throughout the infection period. Results are presented as the mean ± standard deviation (SD) from three independent

experiments. (B) Growth curve of EV-A71 in RD cells treated with 2 μM α-amanitin, 4 μM actinomycin D (ActD), or 100 μM 5,6-dichlorobenzimidazole riboside (DRB). Viral titers were determined by plaque assays. (C) EV-A71 RNA replication in the presence or absence of 2 μM α-amanitin. Relative viral RNA levels were quantified by reverse transcription-quantitative polymerase chain reaction (RT-qPCR). The Y-axis represents the $\log_{10}$-transformed fold change of viral RNA levels at the indicated time points, calculated using ΔCt values. RNA levels were normalized to the amount detected immediately after viral adsorption (time 0), which was defined as 100%. Data in (B) and (C) are presented as the mean±SD from three independent experiments. (D) Upper panel: Schematic representation of the metabolic labeling approach in EV-A71-infected cells. RNAs labeled with 500 μM 4sU at 4 or 8 h post-infection (hpi; MOI 10) were biotinylated and captured using streptavidin beads. Lower panel: A representative RT-PCR result showing the labeling and pull-down efficiency for selected host transcripts and viral RNAs. Data are from at least three independent experiments. (E) Quantitative PCR analysis of the pull-down efficiency for host and viral RNAs. Data represent the mean±SD from three technical replicates of a single independent experiment. (F) Upper: Schematic of the strategy for isolating 4sU-crosslinked RNA-protein complexes (RNPs) from EV-A71-infected cells at 4 or 8 hpi (MOI = 10). UV-irradiated lysates (365 nm) were subjected to a pull-down assay using oligo(dT) beads. Lower: RT-qPCR analysis of the pulldown efficiency using oligo(dT) beads. The RNA levels of viral RNA, host *vimentin* (*VIM*) transcript, and 18S ribosomal RNA (rRNA) pulled down by the oligo(dT) beads were compared with 5% of the input RNA. Data are presented as mean±SD from triplicates of the representative experiment used for mass spectrometry. (G) Representative western blot using the lysates used for mass spectrometry show regulatory proteins known to associate with viral RNAs at different time points. Two percent of the initial lysates used for oligo(dT) pulldown served as the input. Statistical significance was assessed using Student's t-test: ***p < 0.001; ns: not significant. Abbreviations: ActD, actinomycin D; DRB, 5,6-dichlorobenzimidazole 1-β-D-ribofuranoside; vRNA, viral RNA; 4sU, 4-thiouridine; ACTB, β-actin; GAPDH, glyceraldehyde-3-phosphate dehydrogenase; VIM, vimentin; 18S, 18S rRNA.

pull-downs. To address this and enable rigorous analysis, we performed quantitative characterization of temporal RNPs using mass spectrometry (MS).

## Proteomic characterization and pathway enrichment analysis of temporal RNPs post EV-A71 infection

To minimize variability introduced by host shutoff during infection, which could confound comparisons with mock-infected samples, we chose to perform proteomic comparisons between 4sU-labeled and non-labeled samples derived from infected cells processed under identical conditions [19]. Pull-down samples from both the non-4sU and 4sU groups underwent in-gel digestion, and the resulting tryptic peptides were subjected to triplicate liquid chromatography-MS/MS analysis. The average spectral count protein ratios between the 4sU and the non-4sU groups were calculated (a full list is provided in S1 Table). Based on the $\log_2$ mean value (-0.1973 and 0.0375 for the 4-h and 8-h RNPs, respectively) and standard deviation (SD) of total protein ratios, a total of 227 and 225 proteins with ratios exceeding the mean plus one SD were identified as components of the 4-h and 8-h RNPs, respectively (S2 and S3 Tables). After subtracting the 44 proteins detected in both groups, 183 and 181 proteins were specifically detected at 4 and 8 h, respectively (Fig 2A). Consistent with the western blot results (Fig 1F), a protein-protein interaction network analysis of the top 55 proteins defined by two SD above the mean ratio in the 4-h RNPs group revealed the presence of several ITAF proteins, including PTBP1, PTBP2, SRSF3, KHSRP, FUBP1, and FUBP3, as well as the proposed ITAF proteins hnRNPR and SRSF10 (Fig 2B) [17,36,37].

To comprehensively understand the functional properties of the 4-h and 8-h RNP components, we selected proteins with ratios exceeding one SD above the mean ratio and performed Gene Ontology (GO) analysis using the PANTHER classification system [38]. We also examined the protein-protein association networks via STRING database [39]. Enrichment analysis of the 4-h RNPs revealed proteins involved in splicing and translation regulation (Fig 2C), including paraspeckle proteins, members of the serine/arginine-rich (SR) protein family, and hnRNP/PTB proteins. Additionally, nucleolar proteins, as well as cellular and mitochondrial ribosome proteins, were identified in the 4-h RNPs. In contrast, the 8-h RNPs exhibited detection of membrane or lumen proteins from both the endoplasmic reticulum (ER) and mitochondrion, alongside proteins involved in RNA metabolism. Furthermore, membrane-associated proteins participating in vesicle and protein transport were identified (Fig 2D). Proteins detected in both the 4-h and 8-h RNPs were implicated in RNA splicing, mRNA export, and metabolism. Moreover, we identified proteins that bind to the AU-rich element, including the stress granule proteins TIAL1 and TIA1, which are known to regulate EV-A71 replication. (Fig 2E) [16,37,40]. Collectively, these findings suggest that RNPs in EV-A71 infected cells undergo dramatic remodeling from the eclipse to the maturation phase.

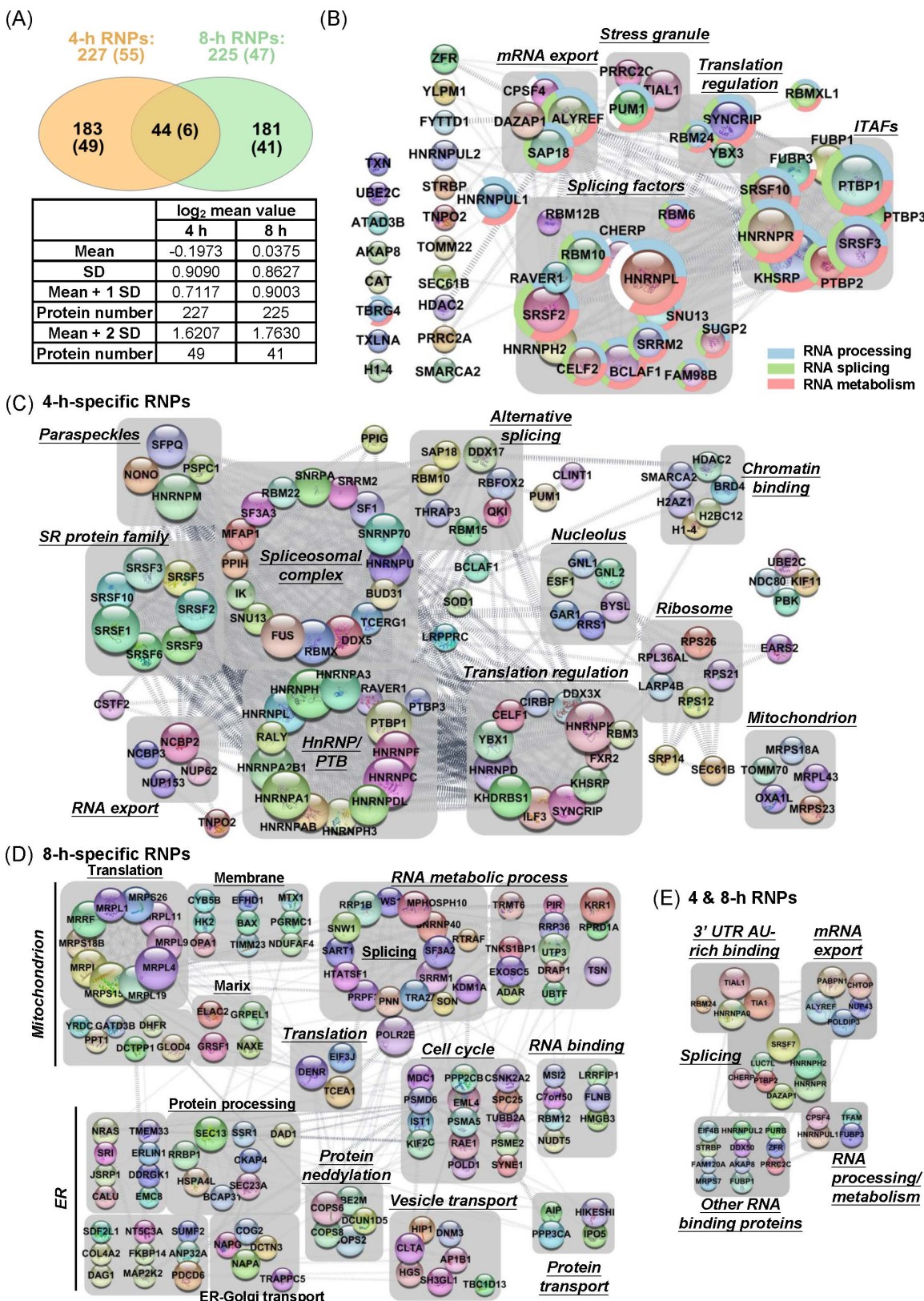

**Fig 2. Gene Ontology and Pathway Analyses of 4-h and 8-h RNPs.** (A) Venn diagram showing the overlap of proteins identified in 4-h and 8-h RNPs using liquid chromatography mass spectrometry (LC-MS/MS). Numbers in parentheses indicate the count of proteins with values ≥2 SD above the

mean. (B) Protein-protein interaction (PPI) network of 4-h RNPs with values ≥2 SD above the mean. Proteins involved in RNA processing, splicing, and metabolism are distinguished by color: blue, green, and pink, respectively. Node size represents the degree of interaction, with larger nodes indicating higher connectivity. Functional complexes are annotated. (C) PPI network of 4-h-specific RNPs with values ≥1 SD above the mean. (D) PPI network of 8-h-specific RNPs with values ≥1 SD above the mean. (E) Proteins consistently detected in RNPs from both 4-h and 8-h time points.

## Cytoplasmic redistribution of phosphorylated SR proteins in early EV-A71 infection

The importance of nuclear proteins in picornavirus replication is well-established [8,37]. However, the high abundance of nuclear proteins detected in the 4-h RNPs may be attributed to the pull-down of host transcripts that evaded α-amanitin inhibition. To validate the association of candidate proteins with vRNAs, we conducted RNA immunoprecipitation (RNA-IP) assays targeting viral transcripts spanning various regions of the EV-A71 genome, including the IRES, P1, P2, and P3 (Fig 3A). Notably, eight proteins (SRSF1–3, SRSF5–7, SRSF9, and SRSF10) out of the 12 SR protein family members were identified in the 4-h RNPs, including the important ITAF SRSF3, which can form cytoplasmic foci involved in translation regulation [11,16,41]. Among these, SRSF7 was detected in both the 4-h and 8-h RNPs, while the remaining seven proteins were exclusively identified in the 4-h RNPs. SR proteins are known to dynamically shuttle between the nucleus and cytoplasm in a phosphorylation-dependent manner, coordinating various RNPs to regulate transcription, splicing, translation, and RNA surveillance. Their roles in both DNA and RNA virus replication are increasingly recognized as significant [42]. Therefore, we pursued their roles in enterovirus infection in the following study. We used the pan-SR antibody (clone 16H3), which targets the arginine/serine (RS)-rich domain found in SR proteins such as SRSF3, SRSF4, SRSF5, and SRSF6, as well as other nuclear RS domain-containing proteins [43], to immunoprecipitate endogenous SR proteins following EV-A71 infection. Specifically, SRSF3, SRSF4, SRSF5, and SRSF6 were detected (Fig 3B). RT-PCR analysis confirmed the successful precipitation of host actin transcripts as controls, along with the detection of viral amplicons corresponding to the IRES, P1, P2, and P3 regions (Fig 3C). Quantitative qPCR analysis further validated the significant association of SR proteins with both host and viral transcripts (Fig 3D), and demonstrated the specificity of our RNA-IP results. These findings highlight the dual role of SR proteins in binding both host and viral transcripts, underscoring their critical function during infection. To further investigate the role of individual SR proteins in EV-A71 replication, we performed knockdown experiments using small interference RNA (siRNA) specifically targeting SRSF3, SRSF4, SRSF5, SRSF6, and SRSF7. The siRNAs downregulated the expression of their target proteins with knockdown efficiencies ranging from 13 to 21% (highest for siSRSF3 at 13.8% and lowest for siSRSF6 at 21.4%) (Fig 3E). Although knockdown of SRSF3 and SRSF5 slightly reduced RD cell viability, no significant cytotoxicity was observed in cells transfected with other siRNAs after 72 h of incubation (S2 Fig). We then performed plaque reduction assays by directly infecting the SR protein-knockdown cells 24 h after siRNA transfection. The results demonstrated a significant decrease in plaque numbers when SRSF4, SRSF5, SRSF6, and SRSF7 were knocked down, compared to the non-transfection mock group (Fig 3F). However, the observed inhibition was moderate, likely due to the incomplete knockdown or potential compensatory effects among SR proteins. Notably, although SRSF4 was not detected in the input samples by mass spectrometry, it was consistently observed as a prominent band in the immunoprecipitates using the pan-SR antibody. Its absence in the proteomic data may be due to low abundance or the physicochemical properties of its peptides, which may hinder detection following trypsin digestion.

Given the association of SR protein hyper-phosphorylation with nuclear import and its role in coordinating various RNA processing complexes, we investigated whether EV-A71 infection affects the expression levels, phosphorylation status, and subcellular localization of SR proteins. To distinguish phosphorylated SR proteins (hereafter referred to as phospho-SR) from their unphosphorylated counterparts, we employed another pan-antibody, clone 1H4 [44–47]. Western blotting analysis revealed dynamic changes in the levels of SR and phospho-SR proteins during infection (Fig 4A). To ensure accurate quantification, all signals were normalized to glyceraldehyde-3-phosphate dehydrogenase (GAPDH), which remained stable across all time points (Fig 4A). At 4 hpi, the levels of SR and phospho-SR proteins

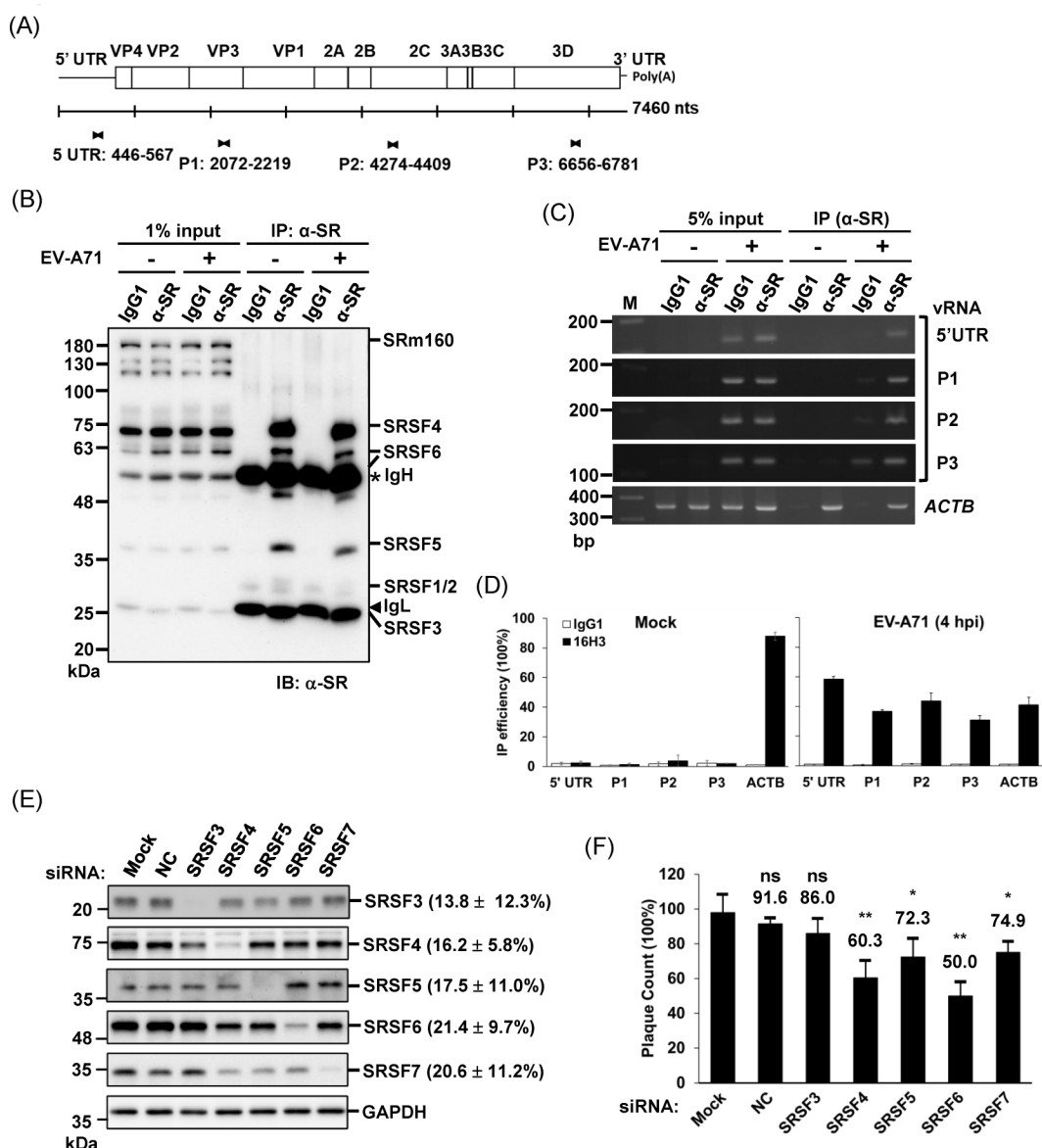

**Fig 3. Validation of the Association Between SR Proteins and Viral RNAs and Their Importance in EV-A71 Replication.** (A) A schematic representation of the viral genome, indicating nucleotide positions of primers used to amplify the 5′ UTR, P1, P2, and P3 regions of the genome. (B) Immunoprecipitation (IP) of SR proteins using the α-SR antibody (clone 16H3). SRSF4 (~75 kDa), SRSF5 (~40 kDa), and SRSF6 (~55 kDa) were efficiently precipitated by this antibody. The results shown are representative of three independent experiments. The star (*) and arrowhead (▾) indicate immunoglobulin heavy chain (IgH) and light chain (IgL), respectively. (C) Representative RT-PCR analysis of immunoprecipitated RNA by the α-SR antibody at 4 hpi of EV-A71 virus. Data shown are consistent with findings from three independent experiments. (D) Quantitative PCR analysis of RNA-IP efficiency for host and viral RNAs. Data are presented as the mean ± SD from three technical replicates of a single independent experiment. (E) Representative western blot analysis of RD cells transfected with siRNAs (100 nM) targeting SRSF3, SRSF4, SRSF5, SRSF6, and SRSF7, respectively. A commercial negative control (NC) siRNA was used as a control. Each SR protein was detected using its corresponding specific antibody. Mock indicates cells without siRNA transfection, and NC refers to the commercial negative control siRNA. Knockdown efficiency of each siRNA is shown in parentheses and is presented as the mean ± SD from three independent experiments. (F) Plaque reduction assay showing the impact of SR protein knockdown on EV-A71 replication. RD cells were transfected with siRNAs for 24 h and then directly infected with EV-A71. After 48 h of incubation, plaques were visualized and counted. Approximately 50–100 plaques were detected in the mock group to ensure statistical robustness. Data represent the mean ± SD from three biological replicates. Statistical analysis was performed using the Student's t-test. **: p < 0.01; *: p < 0.05; ns: not significant.

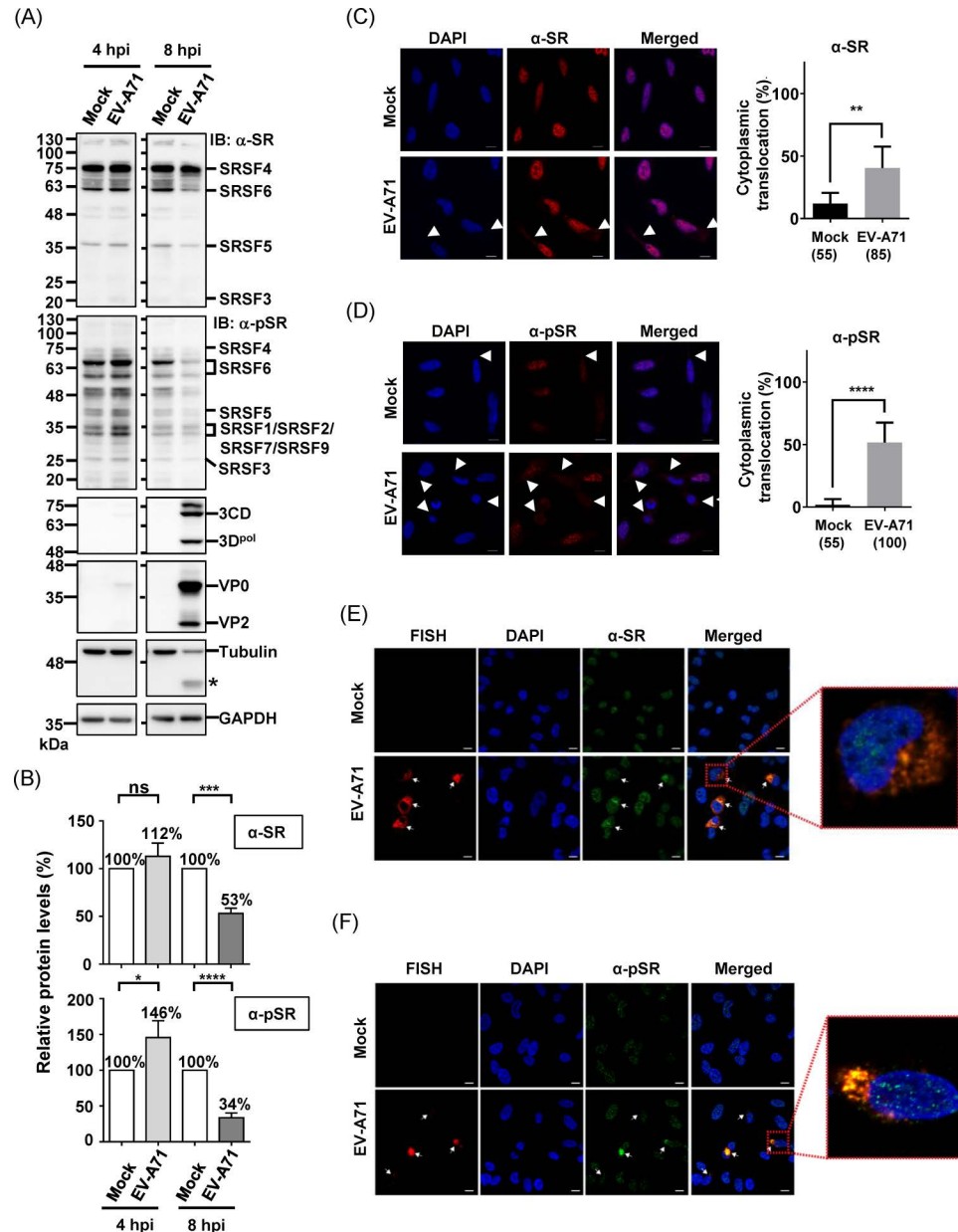

**Fig 4. Alteration of Total Protein and Phosphorylation Levels of SR Proteins and Their Subcellular Distributions during EV-A71 Infection.**
(A) Representative Western blot showing the levels of total SR proteins (detected by α-SR) and phosphorylated SR proteins (phospho-SR, detected by α-pSR) at 4 and 8 hpi with EV-A71. Major SR proteins detected by the pan-antibody, including SRSF4, SRSF5, and SRSF6, and others labeled in the figure, are indicated. Viral proteins were detected using α-3D$^{pol}$ and α-VP0/VP2 antibodies. Tubulin and GAPDH were used as loading controls. The asterisk (*) denotes the cleaved product of tubulin. Data shown are representative of three independent experiments. (B) Quantitative analysis of total and phosphorylated SR proteins detected by α-SR and α-pSR antibodies. Signal intensities were measured from digital images acquired using the ChemiDoc Imaging System (Bio-Rad, Hercules, CA, USA) and normalized to GAPDH. Protein levels in the mock group were set as 100%. Only non-saturated images were used to ensure quantification accuracy. Data are presented as the mean ± SD from three independent experiments. (C) Immuno-fluorescence staining showing the subcellular localization of SR proteins (left panels) in mock and EV-A71-infected RD cells (MOI 40, 4 hpi). Nuclei were counterstained with DAPI. Arrows indicate cells displaying cytoplasmic localization of SR proteins. Scale bar = 10 μm. The bar graph (right panel) shows the cytoplasmic translocation ratio based on counts from randomly selected fields across three independent experiments. Numbers in parentheses indicate the total number of cells analyzed per group. (D) Immunofluorescence staining and quantification of phosphorylated SR proteins as described in (C). Statistical analysis in (C) and (D) was performed using the Student's t-test (**p < 0.01; ****p < 0.0001). (E) Fluorescence in situ hybridization

(FISH) reveals the localization of viral RNAs and their colocalization with SR proteins. (F) FISH analysis shows the colocalization of viral RNAs with phospho-SR proteins. Enlarged images from the indicated regions in (F) and (G) are shown in the rightmost panels for clarity. All fluorescence images, including those for RNA and protein staining, were captured using a confocal laser scanning microscope (LSM780; Zeiss).

increased significantly, reaching 113% and 146% of the mock group, respectively (Fig 4B). S3 Fig provides a detailed breakdown of individual SR proteins, highlighting that several members exhibited elevated phosphorylation (S3A Fig). In particular, SRSF6 showed a marked increase, which is consistent with its strong impact on viral replication observed in our knockdown experiments (S3B Fig). By 8 hpi, although the overall SR protein levels had decreased to 53% of the mock group, phospho-SR proteins showed a more pronounced reduction, declining to 34% (Fig 4A, 8 hpi, and Fig 4B). This highlights a marked shift in the phosphorylation status of SR proteins during infection. Whether this reduction is mediated by phosphatase activity or proteolytic cleavage, as observed with tubulin, remains unclear (Fig 4A). These findings reveal the dynamic regulation of SR protein phosphorylation, which may play a critical role in the host's response to viral infection.

To elucidate the role of phospho-SR proteins, whose steady-state localization is predominantly in the nucleus in uninfected cells, we performed immunofluorescence staining to assess their cellular distribution upon EV-A71 infection. Immunofluorescence analysis revealed that both SR and phospho-SR proteins initially localized to nuclear speckles, characterized by punctate nuclear staining (Fig 4C and 4D, Mock). However, upon EV-A71 infection, a subset of cells exhibited cytoplasmic localization of SR proteins (Fig 4C, α-SR), with a notable increase in cytoplasmic phospho-SR proteins (Fig 4D, α-pSR). Contrary to the conventional belief of nuclear localization of phospho-SR proteins, these findings suggest that EV-A71 infection specifically induces cytoplasmic re-localization of SR proteins, particularly the phosphorylated forms, during early infection. This relocalization was observed only in a subset of cells, consistent with the infection rate under our experimental conditions. To further confirm the association of cytoplasmic SR/phospho-SR proteins with EV-A71 infection, we conducted RNA fluorescence in situ hybridization (FISH) using multiple probes targeting the P1 region. Colocalization of both SR and phospho-SR proteins with EV-A71 RNA was observed in the cytoplasm of infected cells, confirming their association (Fig 4E and 4F, respectively). In summary, our findings provide compelling evidence that in the early stages of EV-A71 infection, SR proteins, particularly in their phosphorylated state, undergo re-localization to the cytoplasm, where they bind to viral RNA.

## Differential distribution of 3D$^{pol}$ and phospho-SR proteins in EV-A71 infection

The colocalization of phospho-SR proteins with viral RNA in the cytoplasm prompted the question of whether these phospho-SR proteins are indeed components of the replication complex. To address this, we conducted organellar fractionation using discontinuous fractionation to isolate fractions enriched with various cellular organelles, including nuclei, mitochondria, ER/plasma membranes, endosomes/lysosomes, Golgi apparatuses, and large protein complexes (spliceosome, ribosome, etc.) (Fig 5) [48]. Using TUFM (Tu translation elongation factor, mitochondrial) and calnexin as markers for mitochondria and ER, respectively, we observed that EV-A71 infection did not alter the overall distribution of cellular organelle at 4 hpi or 8 hpi. Notably, viral proteins 2C and 3D$^{pol}$ were predominantly located in the mitochondria, ER/plasma membrane, and endosome/lysosome at 8 hpi (Fig 5). Phospho-SR proteins exhibited a specific distribution, being present in the fractions containing nuclei, Golgi apparatuses, and large protein complexes, but were barely detected in fractions containing mitochondria, ER/plasma membranes, and endosomes/lysosomes (Fig 5, α-pSR). Interestingly, infection-induced phospho-SR proteins, particularly SRSF4, SRSF5 and SRSF6, were notably enriched in the Golgi apparatuses and large protein complex fractions at 4 hpi, with virus-induced phosphorylation diminished by 8 hpi (Fig 5, EV-A71). By contrast, total SR proteins exhibited a uniform distribution across all fractions, as shown in Fig 5 (α-SR). However, subtle differences were discernible: low molecular weight (MW) SR proteins, such as SRSF4 and SRSF6, were

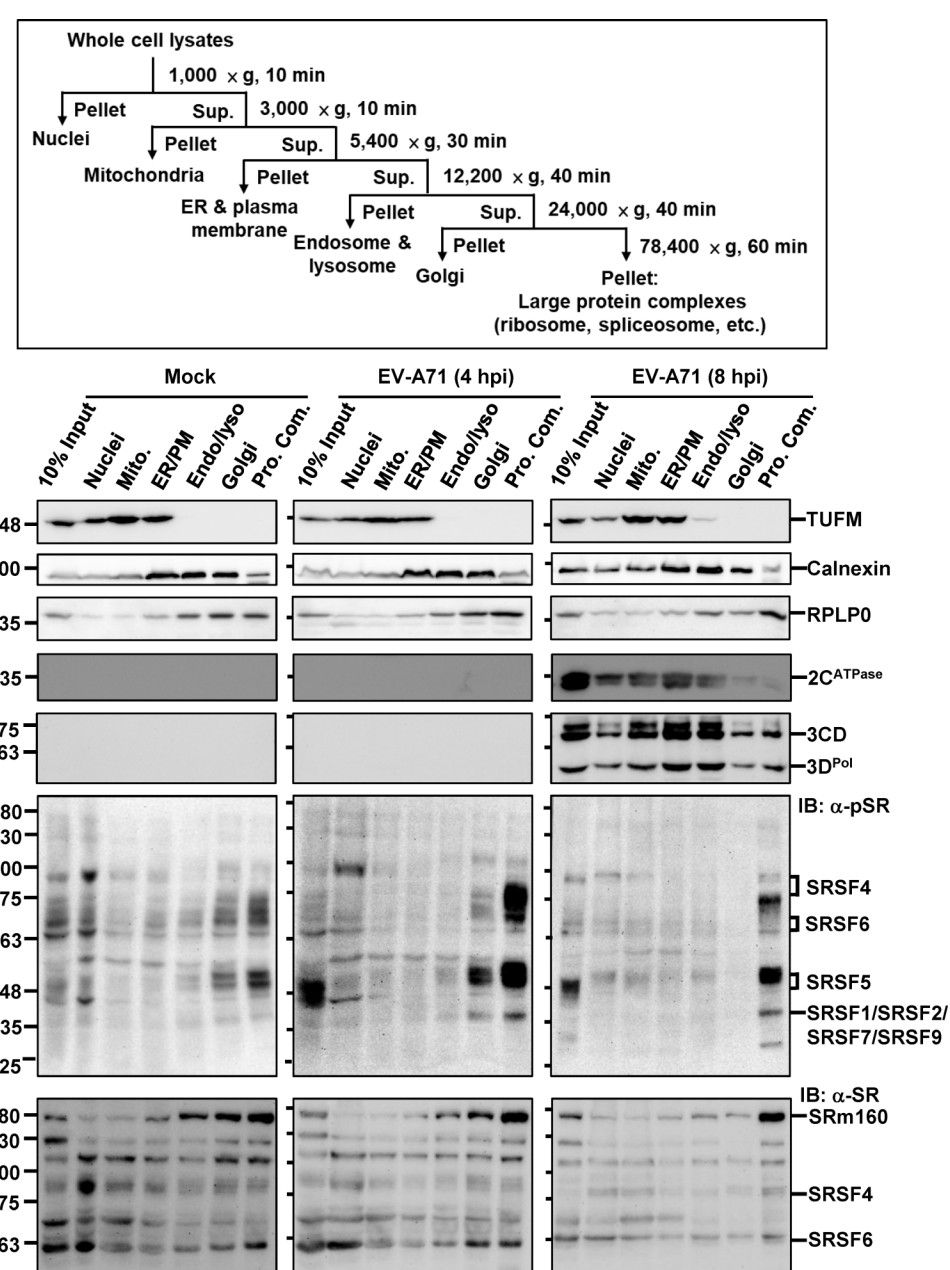

**Fig 5. Enrichment of Phosphorylated SR Proteins in Large Protein Complexes by Assessing Their Distribution Across Cellular Fractions.** The flowchart in the upper panel illustrates the speed and duration used to separate various cellular fractions by discontinuous centrifugation. Pellets from each centrifugation step were resuspended in volumes of sample buffer equivalent to that used for the input, which consisted of a TCA-precipitated pellet from 10% of the initial lysate. Equal volumes of each sample were analyzed by SDS-PAGE and immunoblotting using the indicated antibodies. TUFM and Calnexin serve as markers for mitochondria and the endoplasmic reticulum (ER), respectively. RPLP0 represents the large subunit of ribosomes. Viral proteins at 8 hpi were detected using α-2C and α-3D antibodies. The result is representative of three independent experiments. Abbreviations: Endo: endosome, Lyso: lysosome, Mito.: mitochondria, PM: plasma membrane, Pro. Com.: large protein complexes, RPLP0: Ribosomal Protein Lateral Stalk Subunit P0, RPS6: Ribosomal Protein S6, TUFM: Tu Translation Elongation Factor, Mitochondrial.

primarily detected in the nuclear fraction, while high MW proteins, like SRm160 were predominantly found in fractions containing endosomes/lysosomes, Golgi apparatuses, and large protein complexes. Notably, no obvious differences were observed in the infection group compared to the mock group. Collectively, while we cannot rule out a role for SR proteins in replication, our findings indicate that phospho-SR proteins, including SRSF4, SRSF5, and SRSF6, are largely absent from fractions enriched with viral replicase. Instead, their predominant accumulation in fractions containing large protein complexes suggests a possible association with RNA processing complexes, potentially related to translation or other post-transcriptional events.

## Co-sedimentation of phosphorylated SR proteins with 80S monosomes during the eclipse phase of EV-A71 infection

To investigate the potential involvement of phospho-SR proteins in translation, we conducted polysome profiling of EV-A71-infected cells. Cell lysates from mock-treated or EV-A71-infected cells harvested at 4 hpi and 8 hpi were fractionated using a 10–50% sucrose gradient, generating 16 fractions. At 4 hpi, consistent with studies of other enterovirus infections [41], we observed an accumulated of monosomes due to partial polysome disruption, which became more pronounced at 8 hpi (Fig 6A, upper panel). RT-PCR analysis confirmed a time-dependent shift of host *ACTB* transcripts toward the monosome fraction, while viral RNAs were consistently detected in all fractions enriched with rRNA at both time points (Fig 6A, lower panel). To validate the presence of phospho-SR proteins as part of the ribosome-associated proteins, we treated the infected cells at 4 hpi with puromycin and EDTA, known to disrupt polysomes [49]. While puromycin treatment led to a clear accumulation of monosome fractions in the polysome profiling, EDTA treatment further disrupted the monosomes into 40S and 60S subunits (Fig 6B, upper panel). Both effects were confirmed by the shift of 18S and 28S rRNA into the earlier (lighter) fractions (Fig 6B, lower panel). Western blot analysis revealed that while puromycin treatment did not alter the infection-induced phosphorylation state of SR proteins, EDTA treatment led to a further hyper-phosphorylation (Fig 6C). This effect may result from the inhibition of metal-dependent serine/threonine phosphatases by EDTA [50,51]. In polysome profiling under mock conditions, both SR and phospho-SR proteins were predominantly found in lighter fractions containing free mRNPs and were also partially detected in 40S fractions (Fig 6D). After 4 h of EV-A71 infection, phospho-SR proteins, while still predominantly distributed in free mRNPs, were also detected in fractions containing 40S, 80S, and polysomes. (Fig 6E, CHX). Furthermore, their distribution shifted, showing enrichment in fractions containing 40S and 60S subunits following puromycin and EDTA treatments (Fig 6E, Puro and EDTA). These findings indicate that a portion of phospho-SR proteins are present in the translation initiation complex and are detectable in the polysome fraction during EV-A71 infection. Therefore, EV-A71 infection may redirect nuclear phospho-SR proteins to associate with cytoplasmic viral RNA during the early stages of infection, which might facilitate IRES-dependent translation.

## Inhibition of SR protein phosphorylation impairs RNA virus replication

SR protein phosphorylation, as previously discussed, is crucial for nucleocytoplasmic shuttling and modulating protein–protein and RNA interactions [52]. This process is regulated by multiple kinases: cytoplasmic SR protein kinases (SRPK1–3) initiate nuclear import of SR proteins, while hyper-phosphorylation by nuclear CDC-like kinases (CLK1–4) modulates their activity in pre-mRNA processing. The importance of these kinases has also been demonstrated in the replication of several viruses, including herpes simplex virus (HSV), adenovirus (AdV), human immunodeficiency virus (HIV), and human influenza A virus (IAV) [42,53]. To investigate the role of phospho-SR in the EV-A71 replication through the regulation of IRES-dependent translation, we inhibited SRPK1/2 and CLK1/2/4 using SRPKIN-1 (an irreversible inhibitor) and TG003 (a reversible inhibitor), respectively, and assessed their impact on EV-A71-infected cells [42,54]. The $CC_{50}$ values of SRPKIN-1 and TG003 in RD cells were determined to be greater than 10 μM and 100 μM, respectively. However, prolonged incubation with TG003 resulted in a slight reduction in cell viability (S4A Fig, 72 h). Subsequent experiments

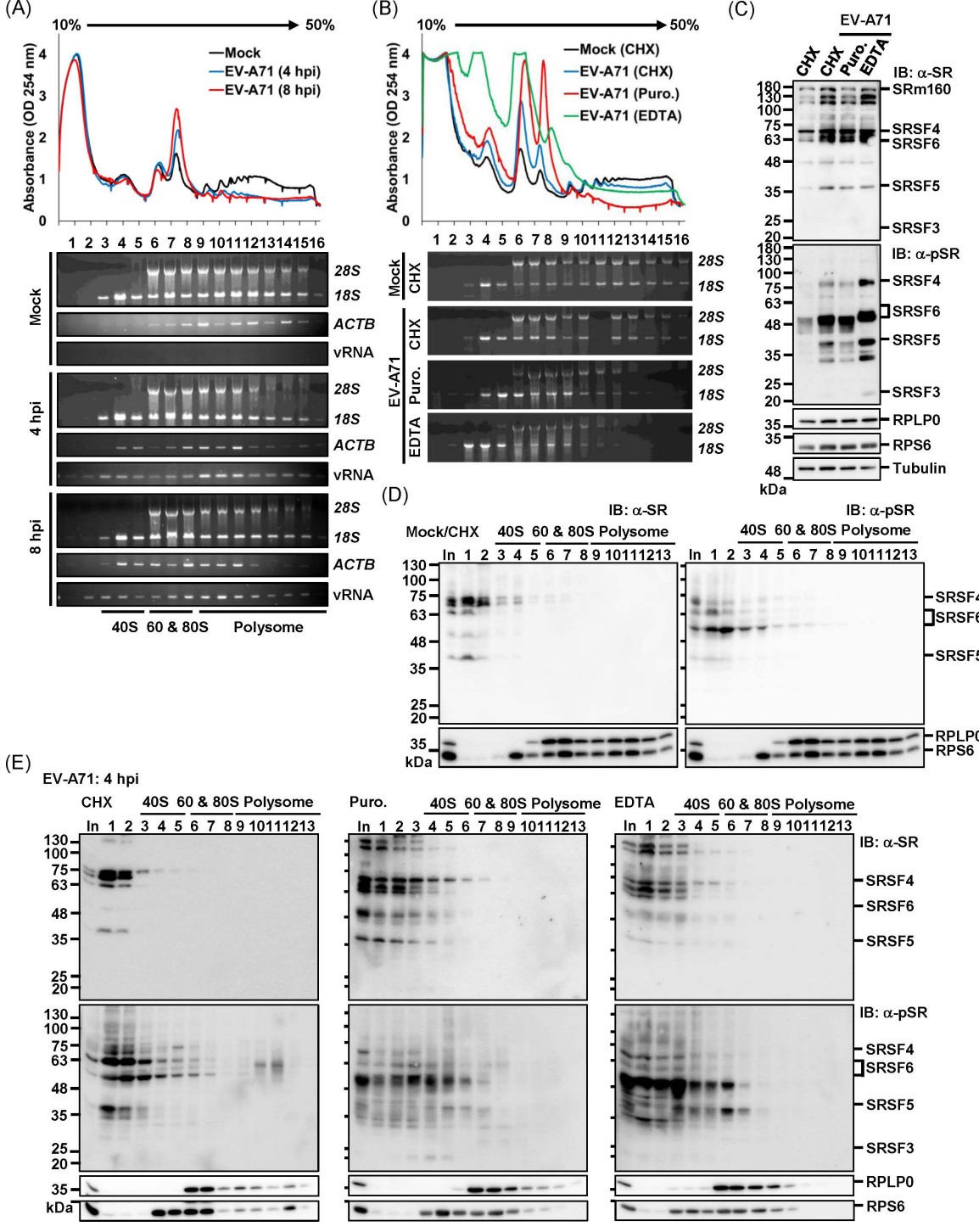

**Fig 6. Co-Sedimentation of the Phosphorylated SR Proteins with Ribosomes at 4 h Post EV-A71 Infection.** (A) Representative polysome profiles of RD cells infected with EV-A71 at an MOI of 10 for 4 h and 8 h are shown in green and red, respectively. Sixteen fractions were collected from the top to the bottom of the sucrose gradient. The distribution of 18S and 28S rRNA, as well as host *ACTB* and viral RNA from the IRES region, is displayed below the profiles. (B) Polysome profiles of EV-A71-infected cells treated with various translation inhibitors are shown in the upper panel, with each drug profile marked by a different color for distinction. The corresponding distribution of 28S and 18S rRNA across fractions is shown in the lower panel. (C) Effects of different translation inhibitors on the phosphorylation status of SR proteins were examined by western blotting. Total SR proteins and phospho-SR

proteins were detected using α-SR and α-pSR antibodies, respectively. Ribosomal proteins RPLP0 and RPS6, along with tubulin, served as controls. (D) Distribution of total SR proteins and phosphorylated SR proteins in uninfected (Mock) cells across gradient fractions. Proteins precipitated by TCA from fractions 1 to 13 were analyzed by western blotting using the indicated antibodies. RPS6 and RPLP0 represent the 40S and 60S ribosomal subunits, respectively. Input (In) represents 10% of the initial lysates used for the experiment. (E) Western blot analysis of total and phosphorylated SR proteins in sucrose gradient fractions from EV-A71-infected cells (4 hpi) treated with cycloheximide (left), puromycin (center), or EDTA (right). All results shown are representative of three independent experiments.

were conducted using 200 nM SRPKIN-1 and 50 µM TG003, which did not cause cell toxicity when combined (S4B Fig). While neither drug alone efficiently inhibited SR protein phosphorylation, western blot analysis revealed notable, substrate-specific reductions (S4C Fig). The most pronounced decrease in phosphorylated SR proteins (approximately 30%) was observed when both inhibitors were used in combination (Fig 7A and 7B, α-pSR). In addition, total SR protein levels were also reduced to approximately 57% under the combined treatment (Fig 7A and 7B, α-SR). Importantly, this combined inhibitor treatment also led to decreased expression of viral proteins VP0 and 3 CD (Fig 7C and 7D). Consistent with this, viral titers were significantly attenuated, reaching 46% of the control, when SR protein phosphorylation was impaired (Fig 7E). To specify the replication stages regulated by phospho-SR proteins, we performed a time-of-addition assay. Cells were treated with inhibitors before infection (denoted by -2 to -1 h), during adsorption (-1 to 0 h), or at the eclipse phase (+4 h) (Fig 7F, upper schematic). Viral titers dropped to approximately 33% of the control when inhibitors were applied throughout the experiment (-2 to +8 h). The inhibitory effect gradually diminished with later drug treatments and was entirely abolished when inhibitors were added at 4 hpi (+4 to +8 h) (Fig 7F), highlighting the critical role of phospho-SR proteins during the early stages of infection.

To evaluate whether the reduced viral replication was due to the inhibition of IRES-dependent translation, we conducted a dual-luciferase assay [55]. RD cells were transfected with a bicistronic reporter plasmid containing or lacking the EV-A71 IRES. Renilla luciferase (Rluc) and Firefly luciferase (Fluc) activities were used to represent cap-dependent and IRES-dependent translation, respectively, with Fluc activity greatly increased in the presence of the EV-A71 IRES [30,56]. To characterize the impact of infection-induced SR protein phosphorylation, cells were infected with EV-A71 at an multiplicity of infection (MOI) of 20 at 5 hpi, treated with different inhibitors, and incubated for an additional 4 hours. Normalization to the DMSO-treated group revealed that TG003, either alone or in combination with SRPKIN-1, significantly reduced IRES-dependent translation (Fig 7H). To mitigate potential interference from host gene expression on reporter plasmid activity, we transfected in vitro-transcribed reporter RNAs and assessed the impact of kinase inhibitors following EV-A71 infection. Consistent with the dual-luciferase assay, luciferase activity was reduced in all drug-treatment groups, further supporting the inhibition of IRES-dependent translation (Fig 7I). We then investigated the impact of SR protein phosphorylation on the replication of other enteroviruses containing picornavirus type I IRES using a plaque reduction assay. Results revealed that all viruses tested herein, including CV-A16, CV-B3, EV-D68 were inhibited by SRPKIN-1/TG003 to varying degrees (Fig 7J).

Beyond enteroviruses, we examined the broader antiviral potential of SRPK/CLK inhibitors. SRPKIN-1/TG003 inhibited human coronaviruses 229E and OC43, whose nucleocapsid protein (NP) contains an SR-rich motif phosphorylated by SRPK1. This phosphorylation is critical for regulating RNP assembly during the viral life cycle [42,57,58]. TG003 also suppressed human influenza A virus (H1N1) by targeting viral mRNA splicing, a process essential for efficient viral protein synthesis (S4D Fig) [59,60]. In addition to RNA viruses, SRPKIN-1 and TG003 inhibited the replication of HSV-1, HSV-2, and human adenovirus 5 by interfering with splicing (S4D Fig) [42,53]. These findings underscore the potential of SRPK/CLK inhibitors as broad-spectrum antiviral agents against both RNA and DNA virus infections. In summary, our study provides evidence that, in addition to their established roles in splicing regulation and RNP assembly, multifunctional phospho-SR proteins contribute to the early phase of enterovirus infection by regulating IRES-dependent translation.

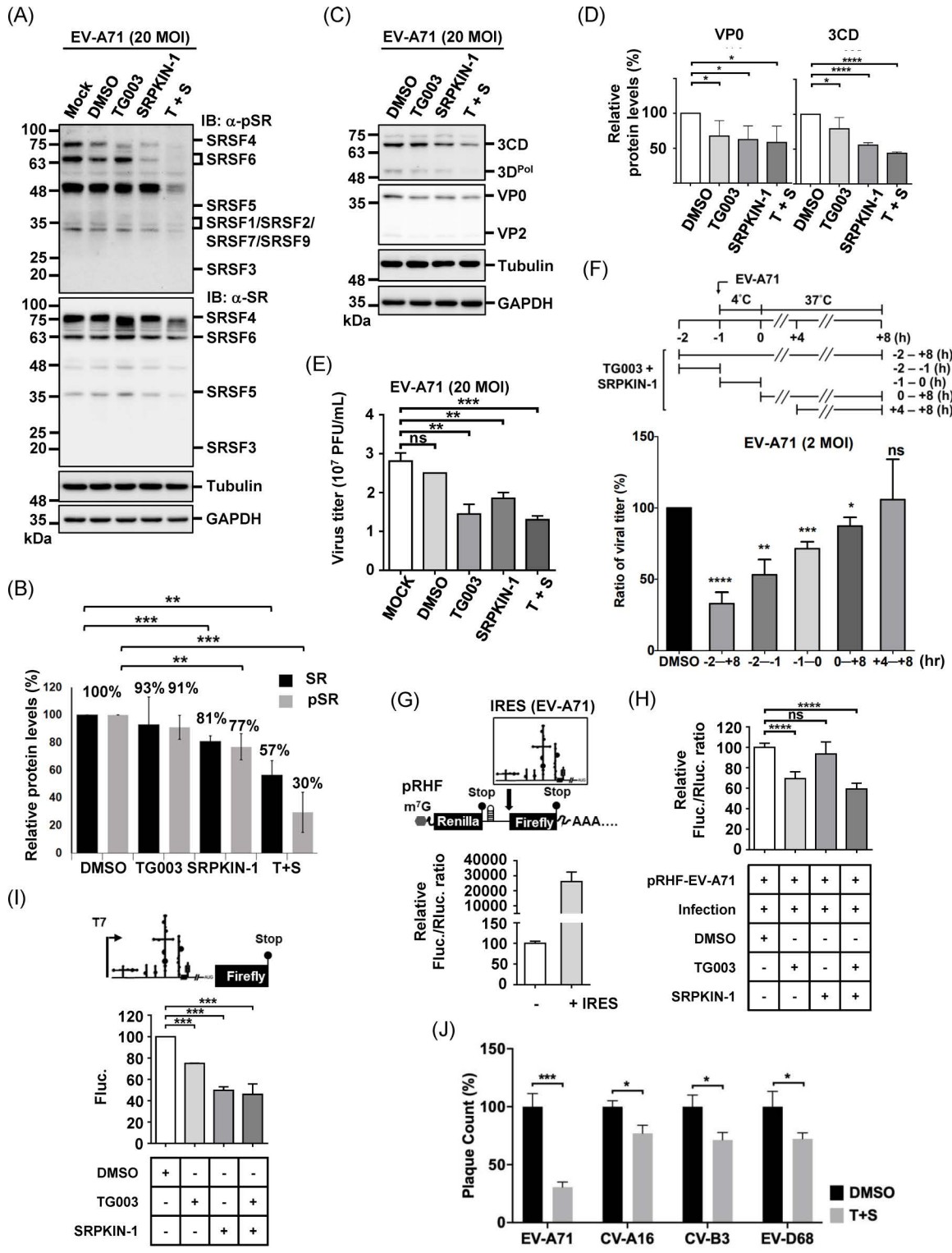

**Fig 7. Inhibition of SR Protein Phosphorylation Impairs IRES-Dependent Translation and Attenuates EV-A71 Replication.** (A) Effects of SR protein kinase inhibitor (SRPKIN-1, 200 nM) and Clk kinase inhibitor (TG003, 50 μM) on SR protein phosphorylation and viral protein expression were analyzed by western blotting using pan-SR (α-SR) and pan-phospho-SR (α-pSR) antibodies. Unless otherwise indicated, the same concentrations were used in all experiments shown. (B) Quantitative analysis of total and phosphorylated SR protein levels following drug treatments. GAPDH was used as

a loading control. Data represent the mean±SD from three independent experiments. Protein levels in the DMSO group were set as 100%. (C) Viral proteins were detected using α-3D$^{pol}$ and α-VP0/VP2 antibodies, with GAPDH as the loading control. (D) Quantification of viral VP0 and 3CD protein levels after drug treatment. Data are presented as the mean±SD from three independent experiments. (E) Viral titers from EV-A71-infected RD cells (MOI 20, 12 hpi) treated with various drugs were determined by plaque assay. Results are shown as the mean±SD from three independent experiments. (F) Schematic representation of the time-of-addition assay (upper panel). TG003 or SRPKIN-1 was added either before or after infection, as indicated. Viral titers were measured at 8 hpi and are shown as the mean±SD from three independent experiments (lower panel). (G) Dual luciferase reporter assay using the pRHF-EV-A71 plasmid, which contains the EV-A71 IRES element positioned between the upstream Renilla luciferase (Rluc) and downstream Firefly luciferase (Fluc) coding sequences (schematic, upper panel). (H) RD cells were transfected with the reporter plasmid for 5 h, followed by EV-A71 infection (MOI 20, 4 h) under the indicated treatments. Relative Fluc/Rluc ratios in (G) and (H) are shown as the mean±SD from a representative triplicate of at least three independent experiments. (I) Mono-luciferase assay using in vitro synthesized RNA containing the EV-A71 IRES upstream of the Fluc CDS (schematic, upper panel). Transfection of in vitro synthesized RNA was performed for 5 h, followed by EV-A71 infection at an MOI of 10 for 4 h under the indicated treatments. The Fluc values are shown as the mean±SD from a representative triplicate of at least three independent experiments. (J) Plaque reduction assay evaluating the effect of SRPKIN-1/TG003 on other enteroviruses, including CV-A16, CV-B3, and EV-D68. Data are shown as the mean±SD from three independent experiments. Statistical analysis was performed using the Student's t-test. ****: p<0.0001; ***: p<0.001; **: p<0.01; *: p<0.05; ns: not significant.

## Discussion

In this study, we conducted RNA interactome analysis of EV-A71 infected-cell using metabolic labeling, coupled with quantitative proteomics, to characterize the composition of temporal RNPs isolated from the eclipse and maturation phases. Our findings reveal dynamic remodeling of RNPs, marked by an initial enrichment of nuclear proteins that were subsequently replaced by cytoplasmic, membrane-bound proteins. Importantly, EV-A71 virus infection induced the phosphorylation and cytoplasmic relocalization of SR proteins, which co-localized with vRNA and co-sedimented with components of the translation apparatus, Functional analyses demonstrated that the knockdown of specific SR proteins, including SRSF4, SRSF5, and SRSF6, significantly attenuated viral replication. Furthermore, pharmacological inhibition of SR protein phosphorylation impaired IRES-dependent translation and reduced viral replication, reinforcing the critical role of SR protein phosphoregulation in EV-A71 replication.

However, several questions remain. While our study underscores the importance of specific SR proteins, the roles of other SR proteins identified in the temporal RNPs during infection remain unexplored. SR proteins may function redundantly or collaboratively, and the extent to which individual SR proteins contribute to viral replication requires further investigation. Additionally, the precise mechanism driving SR protein phosphorylation during early infection remains to be elucidated. The phosphorylation/dephosphorylation cycle of SR proteins could play a pivotal role not only in translation but also in other stages of the viral life cycle, such as replication and assembly, by driving RNP remodeling. For instance, phospho-SR proteins may facilitate protein-protein interactions to form RNPs during early infection. These complexes could then disassemble upon dephosphorylation, allowing for the assembly of new RNPs with other proteins. Furthermore, while many components of the 8-h RNPs are membrane-bound proteins likely involved in replication organelle assembly, their precise roles remain undefined. Together, these dynamics underscore the complexity of RNP remodeling during infection and highlight the need for further investigation.

Despite these limitations, our study provides compelling evidence that EV-A71 exploit host proteins, including phospho-SR proteins, to form distinct functional complexes tailored to their replication needs. Furthermore, the preference of phospho-SR proteins in IRES-dependent translation represents a novel mechanism distinct from host gene expression. While hyperphosphorylated SR proteins are typically localized to the nucleus to mediate pre-mRNA processing, hypophosphorylated shuttling SR proteins escort mRNA to the cytoplasm by interacting with the mRNA export receptor NXF1, thereby activating translation [61–64]. Our findings uniquely highlight the role of phosphorylated SR proteins in cytoplasmic IRES-dependent translation. The multifunctionality of SR proteins, coupled with their phosphorylation-dependent hijacking during infection, underscores the potential of targeting SR kinases as a promising antiviral strategy with broad-spectrum efficacy against RNA viruses [65].

The metabolic labeling has been used to characterize the RNPs composition in cells infected by several viruses, primarily focusing on steady-state RNPs [20–25]. By selectively inhibiting host polymerase activity using α-amanitin, we isolated temporal RNPs from both early and late infection stages (Fig 1F). However, using a Pol II-specific inhibitor alone may introduce noise from transcripts of Pol I and Pol III. To address this, we employed oligo(dT) primer to select poly(A) RNA. However, even with the use of oligo(dT), we cannot exclude the possibility of obtaining nucleolar and ribosomal proteins from labeled rRNA (Fig 2C and 2D). Future studies are necessary to verify their roles in EV-A71 replication. Additionally, given that mRNA and vRNA binding are not mutually exclusive, we focused solely on transcripts specifically labeled at 4 hpi and 8 hpi, without utilizing a mock-infected group as the control. However, due to incomplete transcription inhibition, further validation is required to determine whether the proteins identified in either 4-h or 8-h RNPs represent authentic vRNPs, as demonstrated by the RNA-IP used in this study (Fig 3A–C). Our RNA-IP results clearly show that SR proteins bind to both viral and host transcripts. Additionally, we were able to detect different regions spanning the viral genome, highlighting the effectiveness of metabolic labeling for achieving whole-genome labeling. Knocking down several SR proteins significantly attenuated viral replication, further supporting their critical role in the infection process.

Two other transcription inhibitors, ActD and DRB, were found to inhibit EV-A71 replication. ActD acts by directly chelating into CpG-rich DNA sequences, causing local topological changes that inhibits all three major host polymerases, with the highest efficacy against Pol I [32]. Therefore, ActD can prevent interference from host transcripts and specifically capture proteins interacting with viral RNA. While several viruses, including the poliovirus belonging to *Enterovirus C* species, are capable of replicating upon ActD treatment [20], EV-A71 replication is inhibited even at much lower drug concentration, albeit through an unknown mechanism. Since Pol I inhibition affects ribosome biogenesis, which is crucial in metastatic and cancer cells [66], it remains to be determined whether ActD inhibits EV-A71 replication due to insufficient supply for viral translation demand. The other inhibitor, DRB, targets cyclin dependent kinase 9, which phosphorylates positive transcription elongation factor b, thereby promoting Pol II elongation. Transcription can also be inhibited by targeting casein kinase II (CK2). DRB is characterized by rapid and reversible inhibition, and documented instances of host gene escape from DRB inhibition have been reported [32]. The selective suppression of genes involved in the DNA damage response and p53 pathway, such as sirtuin-1 and CK2a [67,68], suggests that DRB may affect EV-A71 replication by altering specific signaling pathways beyond transcription [69], particularly since CK2 is also involved in SR protein phosphorylation [42].

The results of the time-of-addition assay suggest that while EV-A71 infection increases the phosphorylation levels of SR proteins, pre-existing SR proteins also play a crucial role in the process, with both contributing to the overall phosphorylation during infection (Fig 7F). It is known that SRSF3 export from the nucleus depends on picornavirus protease activity [8]. However, whether the *de novo* phosphorylation occurs remains unclear (Figs 4B and 6C). The regulation of SR protein phosphorylation involves CK2, which acts through SRPK1, and EGF-AKT pathway, which modulates SRPK and CLK activity [42]. It has been observed that both the CK2 and AKT pathways were rewired in SARS-CoV-2 infected cells [70], but whether similar rewiring occurs in EV-A71 infected cells remains to be investigated. In this study, we focused on the SRPK and CLK protein families, comprising SRPK1–3 and CLK1–4, respectively. Currently, there is a lack of potent inhibitor families targeting all members of the SRPK and CLK. Even when cells were treated with SRPKIN-1 and TG003, which inhibit SRPK1/2 and CLK1/2/4 [42,54], complete abolishment of SR protein phosphorylation was not achieved (Fig 7A and 7B). This incomplete inhibition explains the partial decrease observed in viral titers (Fig 7E). When we assessed the impact of kinase inhibitors on viral growth at an MOI of 0.01, significant inhibition was observed only with the combination treatment of SRPKIN-1 and TG003 (S4E Fig). This discrepancy could be attributed to effects on host gene expression caused by the prolonged incubation with inhibitors required for low MOI infections (S4A Fig). While SR proteins are substrates of dual-specificity tyrosine-regulated kinases and pre-mRNA processing 4 [42], their roles in EV-A71-induced SR protein phosphorylation remain unclear. Future research may explore the development of small compounds capable of inhibiting SRPKs, CLKs, and other SR protein kinases.

Through quantitative proteomics, we analyzed RNPs isolated from different phases of EV-A71 replication, identifying several SR proteins in the 4-h RNPs. Polysome profile analyses confirmed the co-fractionation of multiple SR proteins with 80S monosomes and polysomes during early infection (Fig 6E). Knockdown experiments demonstrated the essential roles of SRSF4, SRSF5, SRSF6, and SRSF7 in EV-A71 replication, whereas only a minor, non-significant reduction was observed upon SRSF3 knockdown (Fig 3F). While kinase inhibitors have been widely used to study SR protein-virus interactions [42], it remains unclear whether specific SR proteins coordinate different viral RNA processing steps in a virus-specific manner, as observed in HIV infection [71]. Moreover, given the unique ITAF requirements among different types of picornaviruses, it remains to be determined whether the phosphoregulation of SR proteins also influences the activity of other viral IRES elements, such as those of hepatitis C virus and cricket paralysis virus, which utilize specific tertiary structures to directly interact with eIF3 and the 40S ribosome, respectively [72–74]. This warrants further investigation. In the case of poliovirus, SRSF3 interacts with PCBP2, which directly binds to the stem-loop IV of the IRES, and overexpression of an SRSF3 deletion mutant lacking the RNA recognition motif attenuates poliovirus replication, highlighting the importance of the RS domain in coordinating RNP assembly. Whether the cytoplasmic relocalization and PCBP2 interaction depend on RS domain phosphorylation remains unclear [41]. However, phosphoregulation of SR proteins and other RS-domain-containing proteins likely influences RNA chaperone activity during RNP assembly, as seen with SARS-CoV-2 N protein [58]. Phosphorylation typically increases the overall negative charge of a molecule, potentially impairing RNA-binding activity through electrostatic repulsion. Phosphorylation can also allosterically alter the RS-domain conformation by stabilizing side chain orientations, thereby enhancing RNA/protein recognition. Moreover, phosphorylation can increase SR protein solubility and prevent self-oligomerization [52,75]. This functional flexibility of RS domain is critical for mediating protein–protein and protein–RNA interactions, enabling the assembly of higher-order RNP structures. Such structures involve coordination among SR proteins, RS-domain-containing factors, and diverse RNA processing complexes, facilitating gene expression and protecting RNAs from nuclease degradation [13]. Viral transcripts may exploit cytoplasmic phospho-SR proteins to evade nuclease activity and ensure efficient translation, underscoring the pivotal role of SR protein phosphoregulation in viral replication.

Boersma et al. conducted real-time monitoring of picornavirus translation and replication status at the single-molecule level [6]. Their study revealed that during early infection, vRNAs freely traversed the cytoplasm, whereas at later stages, a significant portion of replicated RNAs aggregated and became immobilized in perinuclear locations [6]. This differential distribution of vRNAs likely reflects RNP remodeling, a process also observed in our study. Notably, we identified several mitochondrial and membrane trafficking proteins in the 8-h-specific RNPs (Fig 2D). Previous research has indicated that EV-A71 infection prompts the re-localization of mitochondria near the nucleus and replication organelles [76]. While EV-A71 replication organelles primarily originate from the ER [77], several mitochondrial proteins have been shown to interact with enterovirus nonstructural proteins [76]. Moreover, EV-A71 infection induces the production of reactive oxygen species associated with mitochondrial morphological anomalies. Given the multifaceted effects of oxidative stress on cellular function, future investigations should explore the impact of mitochondrial protein recruitment during EV-A71 infection [78].

Our findings underscore the significance of RNP remodeling during enterovirus infection, shedding light on how viruses manipulate host phosho-SR proteins to enhance IRES-dependent translation. By integrating our results with advancements in genetic disorders, neurodegenerative diseases, and cancer research, our understanding of the physiological roles of assembling and transitioning various RNPs in gene expression regulation has deepened [65,79]. This exploration of key factors orchestrating essential RNA processing steps offers promising avenues for identifying potential therapeutic targets in the future.

## Materials and methods

### Cell culture and viruses

RD cells (ATCC, No. CCL-136), Madin-Darby Canine Kidney cells (MDCK; ATCC, No. CCL-34) and A549 (ATCC, No. CCL-185) were cultured in Dulbecco's Modified Eagle medium (DMEM; Gibco, Waltham, MA, USA) supplemented with 10% fetal bovine serum (FBS; Gibco), 20 mM L-glutamine, 1 × Antibiotic-Antimycotic (Gibco), and 1 × MEM Non-Essential

Amino Acids Solution (Gibco) at 37°C with 5% $CO_2$. Rhesus monkey kidney cells (LLC-MK2; ATCC, No. CCL-7) and African green monkey kidney cell line (Vero76, clone E6; ATCC, No. CRL-1586), were cultured in Minimum Essential Medium (MEM; Gibco) supplemented as described above. The following virus strains were used: EV-A71 strain MP4 (Accession No. JN544419.1), EV-D68 strain TW-02795–2014 (Accession No. KT711088.1), and coronavirus strains HCoV-OC43 (ATCC No. VR-1558), and HCoV-229E (ATCC No. VR-740). Clinical isolates of Coxsackievirus strains CV-A16 and CV-B3, human influenza A virus H1N1 strain A/TW/3773/2015 (pH1N1), HSV-1 strain 3709/14, HSV-2 strain 2934/14 and adenovirus serotype 5 (Ad5) strain 952/15 were obtained from Linkou Chang Gung Memorial Hospital. All infections were conducted following the guidelines provided by the Taiwan Centers for Disease Control. Enterovirus titers were determined by plaque assay as described previously [80]. LLC-MK2 and Vero-E6 cells were used to propagate HCoV-229E and HCoV-OC43 at 37°C and 33°C, respectively. HCoV plaque assays were performed in RD cells, with 0.5% agarose-containing medium used for HCoV-229E at 37°C and RD cells maintained at 33°C for the HCoV-OC43 infection [81]. Influenza virus pH1N1 was propagated in MDCK cells supplemented with 1 µl 0.25% EDTA-free trypsin per mL. Plaque assays for HSV-1 and HSV-2 were performed in Vero cells using an overlay medium containing 0.2% agarose. For Ad5, plaque assays were conducted in A549 cells with an overlay medium containing 0.5% agarose. To assess the cross-reactivity of host transcription inhibitors, RD cells infected with EV-A71 at an MOI of 10 were treated with various concentrations of α-amanitin (1–2 µM) (Sigma-Aldrich, Inc., St. Louis, MO, USA), ActD (2–4 µM) (Sigma-Aldrich), or DRB (50–100 µM) (Sigma-Aldrich) for 12 h [32]. Viral titers were determined by plaque assay. To inhibit the phosphorylation of SR proteins, cells were treated with 200 nM SRPKIN-1 (MedChemExpress, Monmouth Junction, NJ, USA) and 50 µM TG003 (Sigma-Aldrich), either individually or in combination.

## RNA recovery and RT-qPCR

To purify RNA from cell-based studies, cells were lysed directly using TRIzol Reagent (Invitrogen, Waltham, MA, USA) at the indicated time points, and total RNA was extracted according to the manufacturer's instructions. RNA samples were treated with RQ1 RNase-Free DNase (Promega) at 37°C for 30 min. The DNase digestion was halted by the addition of EGTA (2 mM) followed by heat inactivation at 65°C for 10 min. The resulting DNA-free RNA samples were used as templates for first-strand cDNA synthesis using ReverTra Ace -α- (Toyobo, Osaka, Japan) or SuperScript III Reverse Transcriptase (Invitrogen). Reverse transcription was carried out following the manufacturer's instructions with minor modifications. The RNA and primers were initially denatured at 65°C for 5 min, then incubated with the reverse transcriptase mixture at 42°C for 1 h, followed by a heat inactivation at 95°C for 5 min. Regular PCR and real-time qPCR were performed using 2X SuperRed MasterMix (Toolsbiotech, Taipei, Taiwan) and KAPA SYBR FAST (Roche, Basel, Switzerland), respectively. The primer sequences used to detect host genes are listed below (forward/reverse): actin beta (ACTB): 5′-GCTCGTCGTCGACAACGGCTC-3′/5′-CAAACATGATCTGGGTCATCTTCTC-3′, glyceraldehyde-3-phosphate dehydrogenase (GAPDH): 5′-CCCATGTTCGTCATGGGTGT-3′/5′-GGTCATGAGTCCTTCCACGATA-3′, VIM: 5′-GCAG GATTTCTCTGCCTCTT-3′/5′-GATAACCTGTCCATCTCTAG-3′, and 18S rRNA: 5′- CTCAACACGGGAAACCTCAC -3′/5′- CGCTCCACCAACTAAGAACG -3′. Primers used to amplify different regions of the EV-A71 genome include IRES: 5′-AGTCCTCCGGCCCCTGAATGCGG-3′/5′-GAAACACGGACACCCAAAGTAGTC-3′, P1: 5′-ATGTTCACCGG GTCCTTTATGGC-3′/5′-CTATAATGAGTGTTGCTGATCCATGG-3′, P2: 5′-TCAGCAGCTTCGCAGGAGGAC-3′/5′-CGGT GTTTGCTCTTGAACTGCAT-3′, and P3: 5′-AGTCTCAGCCCAGTGTGGTTCAG-3′/5′-ATAAGTTTTATTGCGGTACACAT GATGGGTGTGA-3′. Regular PCR reactions were performed under the following cycling conditions: initial denaturation at 95°C for 5 min, followed by 25–30 cycles of: 1) denaturation at 95°C for 30 s, 2) annealing at 55°C for 45 s, and 3) extension at 72°C for 1 min. After amplification, a final extension was carried out at 72°C for 7 min. The cycle number for each transcript was determined by testing the input samples, ensuring that a cycle number within the non-saturation stage was selected. For qPCR, reactions were performed on the LightCycler 480 System (Roche) under the following conditions: an initial pre-incubation at 95°C for 2 min, followed by 45 cycles of amplification consisting of denaturation at 95°C for 5 s,

annealing at 60°C for 20 s, and extension at 72°C for 5 s. Subsequently, a melting curve analysis was conducted, beginning with denaturation at 95°C for 5 seconds, followed by annealing at 65°C for 1 minute, and a gradual temperature increase to 97°C to unwind the strands.

## Metabolic labeling

4sU incorporation and RBP isolation were conducted following the protocol outlined by Castello *et al.*, with modifications to enhance incorporation efficiency using 500 µM 4sU (Sigma-Aldrich) [82]. RD cells were infected with EV-A71 at an MOI of 10, with cells treated with α-amanitin (2 µM) prior to and throughout the infection to inhibit host polymerase activity. To assess the labeling efficiency of newly synthesized RNA, total RNA was recovered using TRIzol, as described in the RT-PCR section. Transcripts containing incorporated 4sU were subjected to thiol-specific biotinylation and selected using Streptavidin MagneSphere Paramagnetic Particles (Promega, Madison, WI, USA) as previously described [83]. The selected RNAs were released by treatment with 0.1 M dithiothreitol (DTT, Sigma-Aldrich) and purified using TRIzol LS Reagent (Invitrogen) according to the manufacturer's instructions. To isolate the temporal RNPs at 4 or 8 h hpi, cells were irradiated with 365-nm UV light using a UVP crosslinker CL-1000L (Analytik Jena, Jena, Germany), at 0.2 J cm$^{-2}$ from a ~10 cm distance to crosslink proteins close to the 4sU residues. After irradiation, RNPs were isolated using Oligo(dT)$_{25}$ Magnetic Beads (New England Biolabs, Ipswich, MA, USA) as previously described [82]. Eluates were digested with either RNase A/T1 Mix (Thermo Fisher Scientific, Waltham, MA, USA) for proteomic studies or Proteinase K (Thermo Fisher Scientific) for RT-PCR analysis, according to the manufacturer's instructions. Protein or RNA samples were recovered using 10% trichloroacetic acid (TCA) precipitation or TRIzol LS Reagent (Invitrogen), respectively.

## Western blot

Protein samples were denatured in sodium dodecyl sulfate (SDS)-containing sample buffer (50 mM Tris–HCl, pH 6.8, 100 mM DTT, 2% SDS, 20% glycerol, and 0.2 µg/ml bromophenol blue) at 95°C for 10 minutes. Subsequently, the samples were separated by 8%, 10%, or 12% SDS-polyacrylamide gel electrophoresis (SDS-PAGE) based on the MW of the proteins of interest and transferred to a PVDF membrane (Cytiva, Marlborough, MA, USA). Primary antibodies used for immunoblotting included mouse anti-FBP (clone 6; BD Biosciences, Franklin Lakes, NJ, USA), goat anti-hnRNPI (N20; Santa Cruz Biotechnology, Dallas, Texas, USA), mouse anti-hnRNP C1 + C2/HNRNPC (4F4; Abcam, Cambridge, UK), mouse anti-Enterovirus 71 (clone 422-8D-4C-4D; Sigma-Aldrich), mouse anti-EV 71 3D (clone 4; GeneTex, Irvine, CA, USA), mouse anti-SR (16H3; Sigma-Aldrich), mouse anti-phosphoepitope SR proteins (clone 1H4; Sigma-Aldrich), rabbit anti-GAPDH (Abcam), mouse anti SRSF3 (7B4; Invitrogen), rabbit anti-SRSF4 (Bethyl Laboratories, Montgomery, TX, USA), rabbit anti-SRSF5 (Sigma), rabbit anti-SRSF6 (Bethyl Laboratories), rabbit anti-SRSF7 (Bethyl Laboratories), rabbit anti-RPLP0 (GeneTex), rabbit anti-S6 Ribosomal Protein (5G10; Cell Signaling Technology, Danvers, MA, USA), mouse anti-tubulin (DM1A, Invitrogen), mouse anti-TUFM (Sigma Aldrich), and goat anti-calnexin (Abcam) antibodies. Detection of mouse and rabbit antibodies was accomplished using corresponding Amersham ECL horseradish peroxidase (HRP)-conjugated antibodies (Cytiva), while the goat primary antibody was detected using rabbit anti-goat IgG antibody and HRP conjugate (Sigma-Aldrich). For quantitative analysis of the HRP signal, western blot images were acquired using the ChemiDoc Imaging System (Bio-Rad, Hercules, CA, USA), equipped with a digital CCD camera for chemiluminescent detection. Only images captured without reaching signal saturation were used for quantification to ensure accuracy. Protein band quantification was performed using Image Lab software, version 6.0 (Bio-Rad), following the manufacturer's instructions.

## Proteome analysis

Proteins from both the non-4sU and 4sU groups were separated by SDS-PAGE and stained with colloidal blue (Invitrogen). Subsequently, each gel lane was sliced into 15 fractions, and in-gel digested as previously described [55]. To ensure

robustness, each fraction was divided into three replicates. The proteins in each gel piece were reduced with DTT (10 mM; Biosynth AG, Sankt Gallen, Switzerland), alkylated with iodoacetamide (55 mM; Amersham Biosciences, London, UK), digested with trypsin (20 µg/mL; Promega), and then extracted with acetonitrile. The resulting peptides were identified using MS, as previously described [55]. The MS data have been deposited in ProteomeXchange with the identifier PXD036890 [84].

## Database searching and protein identification

To identify peptides, mass spectra were analyzed using the Mascot algorithm (Version 2.1, Matrix Science, Boston, MA, USA) and searched against the SwissProt human sequence database (released March 2018, selected for Homo sapiens, 20,198 entries) from the European Bioinformatics Institute. The search was conducted using the Proteome Discoverer software (version 2.1, Thermo Fisher Scientific). The peak list was generated using Thermo ExtractMSn software (Version 1.0.0.8, May 2012 release). Parent and fragment ion mass tolerances were set to 10 ppm and 0.5 Da, respectively. Methionine oxidation (+15.99 Da) and cysteine carbamidomethylation (+57 Da) were defined as variable and fixed modifications, respectively. Trypsin digestion was employed, allowing for up to one missed cleavage. A random sequence database was utilized to estimate the false-discovery rates (FDRs) for peptide matches, with the threshold set to < 0.5%. Only proteins with at least two unique peptides were retained for further analysis. A comprehensive list of identified peptides can be found in the S1 Table.

## Spectral counting-based protein quantification

Protein abundance between groups was compared using the label-free spectral counting method to assess protein levels in the non-4sU and 4sU groups [85]. Exclusive spectrum counts for each identified protein were exported using Proteome Discoverer software in Excel (Microsoft, Redmond, WA, USA). To minimize variation between analyses, the normalized spectral count (NSC) for each protein was calculated by dividing the spectral count (SC) of the protein by the total SC of the analysis. Fold changes were determined by dividing the average NSC of the 4sU group by that of the non-4sU group. To avoid division by zero and prevent overestimation of fold changes, missing value (unidentified protein but identified in another sample) was filled in as SC of one. The fold changes in identified proteins were then $\log_2$ transformed and adjusted by global normalization. Log₂ transformation of spectral count ratios was applied to stabilize variance and normalize the distribution of the proteomic data. This approach enables a symmetric evaluation of fold changes and facilitates consistent statistical comparisons across samples. Finally, the $\log_2$ mean values (-0.1973 and 0.0375 for the 4-h and 8-h RNPs, respectively) and SD (0.9090 and 0.8627 for the 4-h and 8-h RNPs, respectively) of the total protein ratios were determined. Proteins with ratios above the mean plus one SD (0.7117 and 0.9003 for the 4-h and 8-h RNPs, respectively) were considered components of the 4-h and 8-h RNPs.

## Protein-protein interaction network analysis

Gene symbols of proteins exhibiting a ratio above the mean plus one or two SD, compared to control samples, were utilized to conduct GO enrichment analyses. This analysis was performed using the PANTHER (Protein Analysis Through Evolutionary Relationships) classification system within The Gene Ontology project [38]. Gene lists associated with GO terms showing the lowest p-values and FDRs, were subject to further analysis for protein–protein interaction networks and functional enrichments using STRING with a confidence cutoff score of 0.7 [39]. The protein network layout was subsequently processed using Cytoscape [86].

## RNA immunoprecipitation

Immunoprecipitation was essentially performed as described by Lee et al. [87], with modifications to accommodate lysates from infected cells. RD cells were infected with EV-A71 at an MOI of 10 for 4 h. Total cells were lysed using hypotonic buffer at 4°C for 15 min, followed by the addition of NaCl to a final concentration of 150 mM. The crude lysates were then

centrifuged at 12,000 rpm for 15 minutes at 4°C, and then the resulting supernatants were collected and stored at -70°C. To precipitate endogenous SR proteins, 1 mg of lysates were mixed with 10 µg of α-SR antibody or a mouse IgG1 isotype control (Abcam) and rotated for 12 h at 4°C. Subsequently, the antibodies were specifically bound by 20 µl of Dynabeads Protein G (Thermo Fisher Scientific) and washed six times with NET-2 buffer containing 0.05% NP-40 (Sigma-Aldrich). While 20% of the immunoprecipitants were saved as a protein control, the remaining were utilized for RNA recovery using TRIzol Reagent, followed by RT-qPCR analysis.

## Plaque reduction assay

To assess the effect of siRNA on viral growth, RD cells were transfected in suspension using Lipofectamine 2000 Transfection Reagent (Invitrogen). Briefly, 0.1 nmol of siRNA and 5 µL of Lipofectamine 2000 were each diluted separately in 250 µL of Opti-MEM I Reduced Serum Medium (Gibco) at 25°C. After 5 min, the diluted siRNA and Lipofectamine 2000 solutions were mixed thoroughly and incubated at 25°C for 20 min. The resulting oligo/Lipofectamine mixture was then combined with $1 \times 10^6$ RD cells in a final volume of 1 mL. The transfection complex was added onto a 3.5 cm dish and incubated at 37°C for 24 h. The siRNA-transfected cells were then infected with the EV-A71 MP4 strain using a virus input calibrated to yield approximately 100 plaques per well after 48 h of incubation, as recommended for robust and countable plaque quantification. Following adsorption, a mixed overlay consisting of 2 mL of DMEM containing 2% FBS and 0.3% agarose was applied, and the cultures were maintained at 37°C for 48 h. Plaques were visualized by fixing cells with 4% formaldehyde, staining with 0.5% crystal violet, and then counting the plaques. The Silencer Select pre-designed siRNAs and negative control (NC) siRNAs used in this study were purchased from Invitrogen. The sequences of the (+) sense strands of the target transcripts are as follows: SRSF3: 5′-GCAACAAGACGGAAUUGGA-3′, SRSF4: 5′-CGCACAGAGUACAGACUUA-3′, SRSF5: 5′-CCACCUGUAAGAACAGAAA-3′, SRSF6: 5′-CAAAUGAGGGUGUAAUUGA-3′, SRSF7: 5′-CUCUCUUCGUAGAUCAAGA-3′. For drug-treated experiments, RD Cells ($1 \times 10^6$) were pretreated with 200 nM SRPKIN-1 and 50 µM TG003 at 37°C for 1 h, and both drugs were maintained at the same concentration throughout the experiment. Infections were performed using a viral dose of 50 PFU/well. After adsorption, a mixed overlay of 2 mL DMEM containing 2% FBS, 0.3% agarose, and SRPKIN-1/TG003 was added to each well. Cells were cultured at 37°C for 36–120 h, depending on plaque size in a virus-specific manner. For HCoV-229E-infected cells, DMEM containing 2% FBS and 0.5% agarose was used as the solidification medium. The cells infected with HCoV-OC43 were maintained at 33°C. MDCK cells were used in pH1N1 infection and cultured in serum-free DMEM containing 0.3% agarose supplemented with 1µl 0.25% EDTA-free trypsin per mL. For HSV-1 and HSV-2, infections were performed in Vero cells with an overlay of DMEM containing 2% FBS and 0.3% agarose. Cultures were fixed at 3 days post-infection for HSV-1 and 2 days post-infection for HSV-2. Ad5 infections were carried out in A549 cells using an overlay of DMEM containing 2% FBS and 0.5% agarose. Infected cultures were fixed at 4 days post-infection.

## Immunofluorescence staining

For immunofluorescence staining of infected cells, $1.2 \times 10^5$ RD cells were seeded onto a coverslip in a 12-well plate and cultured for 24 h. Infection with EV-A71 MP4 viruses at an MOI of 40 was carried out. After 4 h, cells were thoroughly washed with phosphate-buffered saline (PBS) and fixed with 4% paraformaldehyde (Electron Microscopy Science, Hatfield, PA, USA) in PBS at 25°C for 15 minutes. Subsequently, cell permeabilization was performed using 0.5% Triton X-100 at 25°C for 5 minutes. Fixed cells were then blocked with 5% BSA at 25°C for one hour and incubated with primary antibodies 16H3 and 1H4, diluted 1:1000 in PBS containing 5% BSA. These antibodies were detected by using an Alexa Fluor 488-conjugated goat anti-mouse antibody (Invitrogen). The coverslips were mounted with ProLong Gold Antifade Mountant with DNA Stain DAPI (Invitrogen) to simultaneously stain the nuclei with 4',6-diamidino-2-phenylindole (DAPI). Images were captured using a confocal laser scanning microscope (LSM780; Zeiss, Oberkochen, Germany) equipped with a 60 × oil immersion objective. To quantify the percentage of cells with dislocated SR proteins upon EV-A71 infection,

 

more than 50 cells were pooled from multiple fields across at least three independent experiments and analyzed. For RNA in situ hybridization, the RNAscope Multiplex Fluorescent V2 Assay (ACD, Newark, CA, USA) was utilized following the manufacturer's instructions. Specifically, we used RNAscope Probe-V-EV71 (ACD), which targets the MP4 sequence. To co-stain with SR or phospho-SR proteins, RNA detection was performed first, followed by the protein detection as described above. Slides were mounted after completing both RNA and protein detection. Opal 570 fluorophore was employed to distinguish the RNA signal from the α-SR or α-pSR antibody. All fluorescence images for both RNA and protein staining were acquired using the same confocal microscope system as described above.

## Subcellular fractionation and organelle isolation

Discontinuous fractionation was conducted, following the method outlined by Itzhak et al., with some adjustments [48]. Briefly, $1 \times 10^7$ cells were lysed in 4 mL hypotonic lysis buffer (25 mM Tris-HCl, pH 7.5, 50 mM sucrose, 0.5 mM $MgCl_2$, and 0.2 mM EGTA). Lysates were homogenized using a Dounce homogenizer (Kimble, Mainz, Germany) by perfroming 15 strokes at 30-second intervals, followed immediately by adjusting the sucrose concentration from 50 to 250 mM using a hypertonic sucrose buffer (2.5 M sucrose, 25 mM Tris pH 7.5, 0.5 mM $MgCl_2$, and 0.2 mM EGTA). Ten percent of crude cell lysates were precipitated with 10% TCA to serve as the input control. The remaining lysate was subjected to stepwise centrifugation for subcellular fractionation. Nuclei were first pelleted at $1,000 \times g$ for 10 min in a TS-5.1-500 rotor (Beckman Coulter, Allegra 25R). The post-nuclear supernatant was centrifuged at $3,000 \times g$ for 10 min to collect the mitochondrial fraction. Subsequently, the supernatant underwent stepwise centrifugation at $5,400 \times g$ for 30 min, $12,200 \times g$ for 40 min, $24,000 \times g$ for 40 min, and finally $78,400 \times g$ for 60 min in a Type 90 Ti rotor (Beckman Coulter) using an Optima XE ultracentrifuge (Beckman Coulter). All pellets and the input control were resuspended in equal volumes of sample buffer, denatured at 95°C for 10 min, and subjected to SDS-PAGE and immunoblotting.

## Polysome profiling

Polysome profiling of infected cells at an MOI of 10 was performed essentially as described by Lee et al., with the modification that infected cells were incubated in DMEM containing 10% FBS [55]. Briefly, lysates were harvested at 4 or 8 hpi from infected cells grown in three 15 cm dishes. Cells were lysed in 1 mL of lysis buffer containing 20 mM Tris-HCl (pH 8.0), 140 mM KCl, 5 mM $MgCl_2$, 1% (v/v) Triton X-100, 100 µg/mL cycloheximide, $1 \times$ EDTA-free protease inhibitor (Roche), and $1 \times$ PhosSTOP phosphatase inhibitor (Roche). Lysis was performed on ice for 10 min, followed by seven passages through a 26-gauge syringe. The lysates were clarified by centrifugation at $10,000 \times g$ for 20 min at 4°C. Clarified lysates were either used immediately for gradient sedimentation analysis or stored at −70°C. For gradient analysis, 3–4 mg of total protein was layered onto a 10–50% sucrose gradient prepared by diffusion in gradient buffer (20 mM Tris-HCl [pH 8.0], 140 mM KCl, 5 mM $MgCl_2$, and 100 µg/mL cycloheximide). Samples were centrifuged at 39,000 rpm for 2.5 h at 4°C in an SW41 rotor using an Optima XE Ultracentrifuge (Beckman Coulter, Indianapolis, IN, USA). The $OD_{254nm}$ plot and fractionation were performed using a density gradient fractionation system (Teledyne ISCO, Lincoln, NE, USA). Proteins precipitated from 1/3 of each fraction using 10% TCA were analyzed by immunoblotting, while RNA extracted from another 1/3 of each fraction using TRIzol LS Reagent (Invitrogen) was subjected to RT-PCR analysis [55]. Puromycin and EDTA treatments were carried out as described previously [49]. For puromycin treatment, infected cells were incubated with 200 µM puromycin at 37°C for 45 min, followed by rapid cooling and harvesting at 4°C. To dissociate ribosomal subunits using EDTA, lysates prepared as described above were incubated with 30 mM EDTA at 4°C for 10 min. Cycloheximide was omitted from the gradient buffer in both the puromycin and EDTA-treated groups.

## Cell toxicity assay

Cell toxicity was assessed using the CellTiter 96 AQueous One Solution Cell Proliferation Assay (MTS) (Promega). Each drug was prepared as a working solution at $4 \times$ the required concentration and then subjected to a $10 \times$ serial dilution. RD cells were seeded in 96-well plates at a density of $2 \times 10^4$ cells per well and incubated at 37°C for 16 h. The medium was

then replaced with 200 µL of DMEM containing 10% FBS and 50 µL of the drug solution. DMSO was used as the solvent control. After 72 h of incubation at 37°C with 5% $CO_2$, the medium was refreshed with phenol red-free DMEM (Invitrogen) containing 10% MTS reagent. Absorbance at 490 nm was measured using a BioTek Synergy H1 Plate Reader (Agilent Technologies, Santa Clara, CA, USA) to determine the drug concentrations that induced 50% cell death.

### Time-of-addition assay

RD cells were seeded in 12-well plates at a density of $5 \times 10^5$ cells/well and infected with EV-A71 MP4 at an MOI of 2. Cells were treated with 200 nM SRPKIN-1 and 50 µM TG003 at the indicated time points ranging from -2 to +8 h; where 0 indicates the time of virus addition. Adsorption was performed in serum-free medium at 4°C for one hour, after which cells were refreshed with medium containing 2% FBS and incubated at 37°C. Supernatants and lysates were harvested at 8 hpi, and viral titers were determined by plaque assay.

### Luciferase assay

The dual-luciferase assay was performed as described by Lee et al. [55,88]. The bicistronic reporter plasmid pRHF-EV-A71 contains the EV-A71/2231/TW IRES sequence inserted between Renilla and Firefly luciferase coding regions [30,56]. RD cells were seeded at a density of $4 \times 10^5$ cells per well in a 12-well plate and transfected with 0.25 µg of plasmid DNA using Lipofectamine 2000 (Invitrogen), according to the manufacturer's instructions. To assess the effect of infection, EV-A71 was added at a MOI of 20 after 4 h of incubation. Following infection, cells were treated with DMSO, individual inhibitors, or inhibitor combinations and incubated for an additional 4 h. After washing thoroughly with PBS, cells were lysed in 1×Passive Lysis Buffer (Promega) at 25°C for 15 min, followed by centrifugation at 10,000×g for 2 min at 4°C. Renilla and firefly luciferase activities were measured using the Dual-Luciferase Reporter Assay System (Promega) according to the manufacturer's instructions. The raw Rluc and Fluc values are provided in S4 Table. For RNA transfection, 0.25 µg of in vitro transcribed RNA encoding firefly luciferase under the control of the EV-A71 IRES was transfected into RD cells ($4 \times 10^5$ cells per well in a 12-well plate) using Lipofectamine 2000. Transfection, infection, and subsequent steps were performed as described for the dual-luciferase assay, except that only firefly luciferase activity was measured.

### Statistical analyses

Statistical analyses were performed using GraphPad Prism v.10.1.2 for Windows. Results are presented as the mean±SD from at least three independent experiments or biological replicates. For parametric data, including cytoplasmic re-localization of SR proteins, the time-of-addition assay, and the plaque reduction assay, unpaired two-tailed Student's t-tests were used. Analysis of IRES activities and virus growth under various drug treatments was conducted using ANOVA with Dunnett's post-hoc tests. Statistical significance was defined as $P \leq 0.05$.

### Supporting information

**S1 Fig. Inhibitory effects of various transcription inhibitors and RNP pulldown efficiency.** (A) Rhabdomyosarcoma (RD) cells were treated with 4 µM actinomycin D (ActD), 100 µM 5,6-dichlorobenzimidazole 1-β-D-ribofuranoside (DRB), or 2 µM α-amanitin for 4 hours. Reverse transcription-quantitative polymerase chain reaction (RT-qPCR) was performed using total RNA collected at the indicated time points. mRNA levels are expressed relative to the 0-hour time point (defined as 100%) and presented as the mean±standard deviation (SD). Data shown are representative of three biologically independent experiments. (B) Total RNA isolated from RD cells treated with 2 µM α-amanitin, in the presence or absence of 4sU, was subjected to oligo(dT) pulldown at 4 or 8 hours post infection (hpi). RT-qPCR analysis of pulldown samples was normalized to 5% of the corresponding input RNA. Data represent the mean±SD from three technical replicates of a single independent experiment.
(TIF)

**S2 Fig. Cell toxicity of SR protein knockdown in RD cells.** (A) MTS assay evaluating the cytotoxicity of knocking down SR proteins as indicated by the corresponding siRNA of each SR protein. RD cells were transfected with 100 nM siRNA for 24, 48, and 72 hours. Data are presented as the mean ± SD from three technical replicates of an independent experiment. Statistical analysis was performed using the Student's t-test. ***: $p < 0.001$; **: $p < 0.01$; *: $p < 0.05$; ns: not significant.
(TIF)

**S3 Fig. Dynamic changes in SR protein levels and phosphorylation status during EV-A71 infection.** (A) The intensity plot (top panel) shows the migration profiles from mock-infected (red line) and EV-A71-infected (green shaded) RD cell lysates separated by Phos-tag SDS-PAGE. Each peak corresponds to distinct SR protein isoforms, with major proteins (e.g., SRSF1/2, SRSF3, SRSF4, SRSF5, and SRSF7/9) identified based on known migration patterns. Cropped gel images (bottom panels) are aligned with the intensity plot, and blue vertical lines mark the quantified gel regions. Relative mobility (Rf values) is indicated on the x-axis. (B) Quantitative analysis of signals corresponding to phosphorylated SRSF4, SRSF5, and SRSF6 (pSRSF4, pSRSF5, and pSRSF6, respectively). Signal intensities were quantified from digital images acquired with the ChemiDoc Imaging System (Bio-Rad, Hercules, CA, USA). Only non-saturated images were used to ensure accurate quantification. Band intensities were normalized to GAPDH, and protein levels in the mock group were set to 100%. Data are presented as mean ± SD from three independent experiments. Statistical analysis was performed using Student's t-test. ****$p < 0.0001$; **$p < 0.01$; *$p < 0.05$.
(TIF)

**S4 Fig. Cell toxicity of TG003 and SRPKIN-1, and their effects on RNA and DNA virus replication.** (A) MTS assay evaluating the cytotoxicity of TG003 and SRPKIN-1 in RD cells at various concentrations. The 50% cytotoxic concentration ($CC_{50}$) of TG003 and SRPKIN-1 was determined to be ≥ 100 μM and ≥10 μM, respectively. (B) Combinations of TG003 and SRPKIN-1 at different concentrations, based on the $CC_{50}$ values in (A), were tested for their combined effects on RD cell viability. (C) Effects of various inhibitors on the phosphorylation levels of SRSF4, SRSF5, and SRSF6. Protein levels were quantified by normalizing to GAPDH, with the DMSO group set as 100%. Data represent the mean ± SD from three independent experiments. (D) Plaque reduction assays evaluating the antiviral effects of TG003 and SRPKIN-1 against influenza virus H1N1, coronaviruses 229E and OC43, and DNA viruses HSV-1, HSV-2, and adenovirus serotype 5 (Ad5). Data represent the mean ± SD from three independent experiments. (E) Growth curve analysis of EV-A71 replication at low MOI under various treatment conditions, as indicated by the respective symbols. A significant reduction in EV-A71 replication was observed with combined TG003 and SRPKIN-1 treatment (T + S, represented by inverted triangles). Statistical analysis for (B), (C), and (D) was performed using Student's t-test; and for (E) using two-way ANOVA. ****: $p < 0.0001$; ***: $p < 0.001$; **: $p < 0.01$; *: $p < 0.05$; ns: not significant.
(TIF)

**S1 Table. List of proteins identified in the RNP complexes after EV-A71 infection with mass spectrometry.**
(XLSX)

**S2 Table. List of protein components in the RNP complexes 4 h after EV-A71 infection.**
(XLSX)

**S3 Table. List of protein components in the RNP complexes 8 h after EV-A71 infection.**
(XLSX)

**S4 Table. Relative luciferase Activity of firefly and Renilla luciferase of dual luciferase assay.**
(XLSX)

## Acknowledgments

We thank Dr. Bo-Shiun Chen for establishing the organelle fractionation protocol and the Microscopy Center at Chang Gung University for their invaluable technical support with confocal imaging.

**Author's Declaration Regarding AI Assistance:** The authors acknowledge the use of OpenAI's ChatGPT, a large language model, to assist with language editing, grammar correction, and refinement of certain sections of this manuscript. All scientific content, interpretations, and conclusions remain the sole responsibility of the authors.

## Author contributions

**Conceptualization:** Kuo-Ming Lee, Yhu-Chering Huang, Shin-Ru Shih.

**Data curation:** Kuo-Ming Lee, Chih-Ching Wu, Yu-Ting Fan, Huan-Jung Chiang, Pei-Yi Lien, Jui-Ping Wang.

**Formal analysis:** Chih-Ching Wu, Jui-Ping Wang.

**Funding acquisition:** Kuo-Ming Lee, Shin-Ru Shih.

**Investigation:** Kuo-Ming Lee, Huan-Jung Chiang, Pei-Yi Lien, Jui-Ping Wang.

**Methodology:** Kuo-Ming Lee, Chih-Ching Wu, Yu-Ting Fan, Huan-Jung Chiang, Pei-Yi Lien, Jui-Ping Wang.

**Project administration:** Kuo-Ming Lee.

**Supervision:** Kuo-Ming Lee, Yhu-Chering Huang, Shin-Ru Shih.

**Validation:** Kuo-Ming Lee, Chih-Ching Wu, Yu-Ting Fan, Yhu-Chering Huang, Shin-Ru Shih.

**Writing – original draft:** Kuo-Ming Lee, Huan-Jung Chiang, Pei-Yi Lien.

**Writing – review & editing:** Kuo-Ming Lee, Yu-Ting Fan.

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
