## [Decision Letter · Decision Letter 0]

PPATHOGENS-D-24-02812

Subversion of Phosphorylated SR Proteins by Enterovirus A71 in IRES-Dependent Translation Revealed by RNA-Interactome Analysis

PLOS Pathogens

Dear Dr. Lee,

Thank you for submitting your manuscript to PLOS Pathogens. After careful consideration, we feel that it has merit but does not fully meet PLOS Pathogens's publication criteria as it currently stands. Therefore, we invite you to submit a revised version of the manuscript that addresses the points raised during the review process. Especially, address un detail the reviewer's comments made about Figs. 1, 4 and 7.

Please submit your revised manuscript within 60 days Apr 22 2025 11:59PM. If you will need more time than this to complete your revisions, please reply to this message or contact the journal office at plospathogens@plos.org. Please include the following items when submitting your revised manuscript:

We look forward to receiving your revised manuscript.

Kind regards,

Peter Sarnow

Academic Editor

PLOS Pathogens

Alexander Gorbalenya

Section Editor

PLOS Pathogens

 Sumita Bhaduri-McIntosh

Editor-in-Chief

PLOS Pathogens

orcid.org/0000-0003-2946-9497

 Michael Malim

Editor-in-Chief

PLOS Pathogens

orcid.org/0000-0002-7699-2064

**Journal Requirements:**

1) We do not publish any copyright or trademark symbols that usually accompany proprietary names, eg ©,  ®, or TM  (e.g. next to drug or reagent names). Therefore please remove all instances of trademark/copyright symbols throughout the text, including:

- ® on pages: 17, 18, and 23.

3) Please ensure that the funders and grant numbers match between the Financial Disclosure field and the Funding Information tab in your submission form. Note that the funders must be provided in the same order in both places as well. State the initials, alongside each funding source, of each author to receive each grant. For example: "This work was supported by the National Institutes of Health (####### to AM; ###### to CJ) and the National Science Foundation (###### to AM).".

**Reviewers' Comments:**

Reviewer's Responses to Questions

**Part I - Summary**

Reviewer #1: The revised manuscript by Lee et al., has been greatly improved by adding siRNA knockdown of individual SR proteins and performing quantitative analysis of the interactome. Overall, the quality of the figures has been improved. In addition, authors incorporated a detailed description of methods and results, and a more complete discussion on the SR proteins in viral RNA translation and virus multiplication. In my opinion, the conclusions are novel and supported by solid results, despite future work is needed to understand the mechanism of action of all these proteins.

Reviewer #2: Lee at al have submitted an updated version of their paper reporting the RBP interactome of EV71 at different timepoints after infection. The paper is much improved in writing and depth of discussion and details on experiments included.

The paper is still ambitious and uses cutting edge techniques for determining viral RNA interactome at different viral replication stages. The results are potentially novel and could significantly add to the field.

The study uses metabolic labelling of newly synthesized RNA during infection after inhibition of cellular transcription to enrich for viral RNA cross linking to interacting proteins. However the experimental set up does not distinguish between protein hits that bind to viral RNA or residual host mRNA.

The authors follow up on the SR family of proteins identified and perform experiments to test their impact on EV71 replication.

Reviewer #3: The manuscript by Lee et al., that identified SR proteins by metabolic labeling has been revised to address the reviewer comments. Overall the authors have been more forthcoming about the limitations of their studies. This means that the short comings of the study are still present for the most part but now the reader will at least understand what they are. There are still significant issues with how they present their data and how well that is explained to the reader. Specifically, figure 1 shows that they have optimized some of their conditions for the pulldown such as the transcriptional inhibitor, but not others such as how clean the 4sU labeling and pull-down is. Given the preliminary data in figure 1, it seems clear that there will be a lot of non-specific proteins identified using their approach. Indeed, looking at Table S1 there are a lot of proteins that were identified. Curiously figure 2B shows the proteins that were at least two standard deviations above the mean suggesting these were the most vRNA associated proteins, but they didn’t really explore these, they went straight to the SRSF family of proteins, most of which were 1 standard deviation above the mean. Also, SRSF4 which was not identified by proteomics had a bigger effect on viral titers upon knockdown than most of the ones that were identified by mass spec. While one would not expect to have large effects on viral titers by knocking down a single RNA binding protein, SRSF3 had no affect while SRSF5 and 7 were only about 25% down, with SRSF4 and SRSF6 knockdown reducing titers by 40-50%. Also, it is noted that no cell viability data or control experiments showing that knockdown of these host proteins did not make the host cell less fit for viral amplification. The remainder of the manuscript focuses on finding a role for these SR or phospho-SR proteins. They look at co-localization with viral RNA, cell fractionation studies, polysome association, and IRES-mediated translation for which they observed a <2-fold change. This is on par with the KD experiments and similar to what others have seen when knocking down an RNA binding protein and performing either translation or plaque assays. Obviously, the authors have done a lot of new experiments to strengthen the manuscript since the initial review. The biggest issues are whether these host factors globally affect cell fitness for viral replication. This was not tested by using an unrelated virus such as a DNA virus. There is some worry that the claim of a pan antiviral target may just be due to inhibiting these proteins affects cell fitness for viral replication. The authors clearly worked to address the reviewers’ comments, however, there are still a number of issues with their experiments, experimental approaches, presentation of the data and conclusions that somehow leave the reader unsatisfied with their findings.

**Part II – Major Issues: Key Experiments Required for Acceptance**

Reviewer #1: -

Reviewer #2: A major issue affecting the robustness of the papers conclusions is still quantification of protein and phospho protein levels. This is particularly affecting conclusions in figure 4A where no loading control is included in A or B, while tubulin is shown in panel C (4hrs) but is cleaved at 8 hours. If Tubulin is not a suitable loading control due to its cleavage, why not use a different loading control. Although the authors have changed their analysis procedure for quantifying gels, the small changes of 10-20 % are difficult to interpret on the gels shown and the relevance of such small changes is not clear. It looks like all signals in the gels change in a similar direction whether they are labelled as SR proteins or are non specific proteins.

Of more concern, and I may be missing something here, is Figure 7A and B. The authors use drug treatments to inhibit phosphorylation of SR proteins and conclude that the phosphorylation state impacts virus replication and IRES activity. But in 7 A and B the decrease in the phosphorylated SR protein signal (IB: anti pSR) appears to correlate in both the gel and the quantification panel with the decrease in the total protein (IB: anti-SR) suggesting that the ratio of phospho to non phospho SR proteins (which is not shown but should be) does not change with the drug treatment. If this is the case then the conclusions and discussions on this aspect of the paper are not substantiated.

A second issue is still the lack of a mock infected control to try and determine which hits are due to interaction with viral or host RNA that is still present due to purification by oligodT beads.

Reviewer #3: Major comments:

In Figure 1F, it is unclear how much of the oligo(dT) and input are loaded on to the protein gel and how to think about this comparison. One would expect to see enhancement when 4sU was added for the oligo(dT) pull-downs but this doesn’t appear to be true for the 3CD and 3Dpol. The idea should be that leaving out the 4sU should not crosslink proteins so this would be the background. The viral proteins appear to be either the same or higher levels in the -4sU suggesting that the oligo(dT) pull-down is not clean.

Lines 185-6: “In contrast, proteins involved in genome replication were detected in the 8-h RNPs including the 3Dpol and hnRNPC (Figure 1F, lanes 6 and 8)” This statement doesn’t support what is observed Fig 1F. hnRNPC1/C2 is slightly higher with 4sU than without but this is very slight. What is more striking is how high it is in the 4 hpi with 4sU, which is not mentioned.

Figure 1G: “RT-PCR analysis of the pull-down efficiency for RNA-protein complexes. Data show

consistent results across at least three independent experiments.” Is this what should be expected? It seems to this reviewer that if oligo(dT) selected polyadenylated RNAs that there should have been increased ActB and vRNA in the oligo(dT) lanes compared to the input lanes which should be mostly rRNA. Where are the “at least three independent experiments?”. Lines 180-181: “RT-PCR analysis confirmed a comparable level of viral and host transcripts being

pulled down (Figure 1G).” This review would expect that proof that the approach was working should result in more polyadenylated RNA in the pull-downs and less rRNA, which wasn’t shown.

Lines 194-6: “The average spectral count protein ratios between the 4sU and the non-4sU groups were calculated (a full list is provided in supplemental table S1). Based on the log2 mean value (-0.1973 and 0.0375 for the 4-h and 8-h RNPs, respectively)”. What is the purpose of the Log2 transformation? It seems to this reader to alter the data from being intuitive to something that is not. Why not just report the average spectral count ratios of the 4sU to the control. This would provide numbers above >1 as being enhanced and those at or below as not being enhanced. Then it’s just a matter of determining a reasonable cut off.

Lines 311-313, the authors conclude from the cell fractionation and western analysis in figure 5 that: “Collectively, while the possibility of SR proteins regulating replication cannot be excluded, these findings suggest that RNPs enriched with phospho-SR proteins, including SRSF4, SRSF5, and SRSF6, may play significant roles beyond replication, potentially in processes such as translation.” The authors are correct they can’t rule out anything, but this reviewer has difficulty with the rest of the sentence as this data in figure 5 is not well controlled for the amount of protein loaded in each gel nor is there any significant changes that can be determined from these westerns. Also, the golgi lane for the pSR western seems to not be loaded equally as the background bands are noticeable fainter. Regardless this data would not even in its best state be able to speak as to whether these proteins play significant roles beyond replication… or in translation. Rather this conclusion seems to appropriately capture the data in figure 5 (line 310) “Notably, no obvious differences were observed in the infection group compared to the mock group.” Indeed, there is nothing striking presented in this data to shed further light on what, if any, role the SR or pSR proteins are doing during EVA71 infection.

Figure 6C and lines 330-331 the authors conclude: “Western blot analysis revealed that the infection-specific hyper-phosphorylation state of SR proteins remained unaffected by either treatment (Figure 6C).” However, in the pSR EDTA lane, the banding pattern is distinctly different than in the other lanes.

Lines 340-341” The authors state: “suggesting that the virus may redirect nuclear phospho-SR proteins to regulate translation during the early stages of infection.” The authors may want to be careful about over interpreting their findings as there are many interpretations for why the phosphor-SR proteins are polysome associated including that they are bound to the vRNA but not necessarily regulating its translation.

Figure 7D. It seems to this reviewer that a couple of things could be occurring. 1) that the pSR proteins somehow affect viral entry or the antiviral response but clearly the longer the inhibitors are present the larger the effect. It would have been nice to also see if washing out the drug would allow the authors to point to a specific window when the pSR proteins were required. In this way they could keep the amount time the cells were treated the same while probing specific times before, early or late in infection the drug inhibited titers. Also, showing that an unrelated virus, such as HSV or Ad, could replicate in the presence of this drug would be great. Cell viability assays can be less sensitive and problematic if they are not performed during the linear range.

Figure 7E. First the luciferase assays should not be normalized to a negative control, pRHF. Rather it should be normalized to a positive control such as pRHF-EV-A71 with viral infection set to 100%. This avoids issues with insignificant changes in the negative control having a profound effect on the significant numbers. Which strain of EVA71 was included in the reporter? Some have been shown to have cryptic promoter activity and therefore wouldn’t work in a DNA reporter. Have all the controls been performed for this plasmid? If so, include references as this data is not included. Figure S4 reveals the raw RLuc and Fluc values, however, ideally the raw Fluc values should look like the normalized data, but it does not rather the raw Fluc data appears to be unchanged from DMSO to the T+S drug combination. This would suggest that the change in the ratio is more due to changes in cap-dependent translation. It would have been more helpful to present the raw data for the n>3 biological replicates rather than a bar graph a single biological replicate with 3 technical replicates.

The authors should comment on why they did not see SRSF4 in their initial proteomics. Also, perhaps it was there if they looked a back at the data. If not, then why?

DNA and RNA transfections were not described in the methods.

**Part III – Minor Issues: Editorial and Data Presentation Modifications**

Reviewer #1: -

Reviewer #2: 1. Authors should read through the paper carefully as there are multiple spelling errors that should be fixed.

2. The use of Reduction(%) in the axes in figures 7B and Supp Fig 2 and 3 should be replaced as it is not clear what is meant, and the legend does not clarify.

Reviewer #3: Minor comments

Figure 1E, I have no idea what the authors mean by pull-down efficiency.

Figure 1 legend needs to be specific about when the drug was added relative to the infection being performed. This is in Figures 1D,F but needs to be sooner in the legend assuming they were done the same. Also, it is noted that Figure S1 did not show a concentration range of ActD or the other inhibitors to determine optimal drug concentrations for each inhibitor. Also figure S1 used 4µM ActD while figure 1 used 5µM ActD.

Figure 1B, the authors normalize the viral RNA to ActB transcript levels which seems odd given that Figure S1 shows ActB mRNA levels are decreasing with drug treatment as expected. One might expect the authors to normalize to an RNA that is not changing such as rRNA levels. Are the vRNA levels going up or is the ActB levels going down or both? Also, what does the Y-axis label mean Log10 100%?

Figure 3E the authors need to specify the antibody or antibodies used.

Figure 3F what is mock? Is this no siRNA control? What is NC? These need to be defined in the legend. What time post infection were the plaque assays done? Also, was the virus harvested and then plaqued on fresh cells no knockdown or were the plaque assays performed on cells that were knocked down? This needs to be clear how the assay was done. If mock is no siRNA transfected it is a nice control as siRNA KD surely affects the interferon response of a cell.

Figure 4D It is not at high enough magnification or the figure is not at high enough resolution to see nuclear speckles. Also the cytoplasmic localization of the SR proteins is not clear at this magnification. However, the pSR cytoplasmic localization following infection is clear.

Figure 4E: is this still at an MOI of 40 like was done in figure 4D? If so wouldn’t one expect the SR proteins to relocalize to the replication complexes more than half of the time?

Figure 4G, was this confocal microscopy? This needs to be clarified.

Figure 7H should be re-labeled 7G.

The authors need to clarify in the cell viability assays what the working stock solutions were made in, DMSO or media? It would be useful to know what %DMSO was in the final well with the cells. Also, how long after the MTS reagent was the absorbance read? Do the authors know whether this is in the linear range of the assay?

PLOS authors have the option to publish the peer review history of their article (what does this mean? ). If published, this will include your full peer review and any attached files.

**Do you want your identity to be public for this peer review?** For information about this choice, including consent withdrawal, please see our Privacy Policy .

Reviewer #1: No

Reviewer #2: No

Reviewer #3: No

**Figure resubmission:**
---

## [Decision Letter · Decision Letter 1]

PPATHOGENS-D-24-02812R1

Subversion of Phosphorylated SR Proteins by Enterovirus A71 in IRES-Dependent Translation Revealed by RNA-Interactome Analysis

PLOS Pathogens

Dear Dr. Lee,

Thank you for submitting your manuscript to PLOS Pathogens. We are happy to accept your revision.

Yours sincerely,

Peter Sarnow

Academic Editor

PLOS Pathogens

Alexander Gorbalenya

Section Editor

PLOS Pathogens

Michael Malim

Editor-in-Chief

PLOS Pathogens

orcid.org/0000-0002-7699-2064
---

## [Editor Report · Acceptance letter]

Dear Ph.D. Lee,

We are delighted to inform you that your manuscript, " Subversion of Phosphorylated SR Proteins by Enterovirus A71 in IRES-Dependent Translation Revealed by RNA-Interactome Analysis ," has been formally accepted for publication in PLOS Pathogens.

Best regards,

Sumita Bhaduri-McIntosh

Editor-in-Chief

PLOS Pathogens

orcid.org/0000-0003-2946-9497

Michael Malim

Editor-in-Chief

PLOS Pathogens

orcid.org/0000-0002-7699-2064